# Extracellular mRNA transported to the nucleus exerts translation-independent function

Takeshi Tomita [1,2], Masayoshi Kato [1,2], Taishi Mishima[3], Yuta Matsunaga[3], Hideki Sanjo[4], Ken-ichi Ito[5], Kentaro Minagawa[6], Toshimitsu Matsui [7], Hiroyuki Oikawa [8], Satoshi Takahashi[8], Toshifumi Takao[9], Noriki Iwai[10], Takashi Mino [10], Osamu Takeuchi [10], Yoshiro Maru [3 ✉] & Sachie Hiratsuka [1,2 ✉]

RNA in extracellular vesicles (EVs) are uptaken by cells, where they regulate fundamental cellular functions. EV-derived mRNA in recipient cells can be translated. However, it is still elusive whether "naked nonvesicular extracellular mRNA" (nex-mRNA) that are not packed in EVs can be uptaken by cells and, if so, whether they have any functions in recipient cells. Here, we show the entrance of nex-mRNA in the nucleus, where they exert a translation-independent function. Human nex-*interleukin-1β* (*IL1β*)-mRNA outside cells proved to be captured by RNA-binding zinc finger CCCH domain containing protein 12D (ZC3H12D)-expressing human natural killer (NK) cells. ZC3H12D recruited to the cell membrane binds to the 3'-untranslated region of nex-*IL1β*-mRNA and transports it to the nucleus. The nex-*IL1β*-mRNA in the NK cell nucleus upregulates antiapoptotic gene expression, migration activity, and interferon-γ production, leading to the killing of cancer cells and antimetastasis in mice. These results implicate the diverse actions of mRNA.

[1] Institute for Biomedical Sciences, Interdisciplinary Cluster for Cutting Edge Research, Shinshu University, School of Medicine, Matsumoto, Nagano, Japan. [2] Department of Biochemistry and Molecular Biology, Shinshu University, School of Medicine, Matsumoto, Nagano, Japan. [3] Department of Pharmacology, Tokyo Women's Medical University, Shinjuku, Tokyo, Japan. [4] Department of Molecular and Cellular Immunology, Shinshu University, School of Medicine, Matsumoto, Nagano, Japan. [5] Division of Breast, Endocrine and Respiratory Surgery, Department of Surgery, Shinshu University, School of Medicine, Matsumoto, Nagano, Japan. [6] Department of Hematology/Oncology, Penn State College of Medicine, Hershey, PA, USA. [7] Department of Hematology, Nishiwaki Municipal Hospital, Nishiwaki, Hyogo, Japan. [8] Institute of Multidisciplinary Research for Advanced Materials, Tohoku University, Sendai, Japan. [9] Institute for Protein Research, Osaka University, Suita, Osaka, Japan. [10] Department of Medical Chemistry, Graduate School of Medicine, Kyoto University, Kyoto, Japan. ✉email: maru.yoshiro@twmu.ac.jp; hira@shinshu-u.ac.jp

Endogenous mRNA in cells is recognized as a key molecule in the central dogma of biology. Extensive research has been conducted to comprehend the structure and function of mRNA, and the consensus is that mRNA is translated into protein in the cytoplasm[1]. In addition to endogenous mRNA, the presence of extracellular mRNA in extracellular vesicles (EVs), such as microvesicles and exosomes, has been reported[2–6]. EV-derived mRNA can be transferred to other cells, where these are translated into corresponding proteins in vitro[3,4] and in vivo[5,6]. Remarkably, EV-containing mRNA from human endothelial progenitor cells is transferred to quiescent endothelial cells (ECs) and activates an angiogenic program[4]. The observation suggests a functional significance of mRNA in EVs for angiogenesis. The potential significance of mRNA in EVs has also been suggested in the field of tumor biology because tumor-derived EVs contain mRNA for growth factors that could be transferred into monocytes[2]. However, it has not been explored whether "naked nonvesicular extracellular mRNA" (nex-mRNA) outside cells that are not packed into EVs can be transferred into cells and, if so, whether such nex-mRNA in recipient cells function in a translation-dependent fashion.

Tumor metastasis is one of the most important issues in which intercellular communications are involved. Many valuable studies have revealed that metastasis involves complex systemic events between host cells and tumor cells[7–10]. In particular, the host resident tissue and immune cells create a premetastatic phase, with a tumor-friendly environment, before tumor cells physically appear in the distant organ[7,11–15]. Potential antimetastatic immune cells in the premetastatic soil were recently discovered[16]. Specifically, B220+CD11c+NK1.1+ natural killer (NK) cells were found relocated from the liver to the lungs of tumor-bearing mice. These cells accumulate in fibrinogen+-hyperpermeability regions[17] and demonstrate two antimetastatic activities: they eliminate enriched fibrinogen and possess tumoricidal abilities with interferon-γ (IFN-γ) production.

In this study, a comparison of gene expression profiling between B220+CD11c+NK1.1+NK cells in the liver and lungs is carried out to elucidate the molecular mechanisms that underlie the relocation of B220+CD11c+NK1.1+NK cells from the liver to the lungs and their antimetastatic abilities. The profiling indicates the highest fold change of mRNA for zinc finger (ZF) CCCH domain containing protein 12D (ZC3H12D), a putative tumor suppressor in lymphoma and lung cancer patients[18,19], in lung cells (Supplementary Table S1, GSE76235). ZC3H12D belongs to an RNA-binding ZF family, and interaction between ZC3H12D and a nex-mRNA is found. Subsequent analyses reveal that ZC3H12D carries the nex-mRNA to the nucleus where the 3′-untranslated region (UTR) of the nex-mRNA exerts translation-independent antimetastatic functions in vitro and in vivo.

## Results

**An RNA-binding protein located on the cell surface**. To determine whether ZC3H12D+ leukocyte mobilization was induced by primary tumors, immunohistochemical (IHC) examination of the lungs of tumor-bearing mice was conducted. Their tumors originated from E0771 breast cancer cells or Lewis lung carcinoma (LLC) cells. The number of ZC3H12D+CD45+ cells in the lungs increased with primary tumor growth (Supplementary Fig. S1a: E0771 data). Zc3h12d expression in leukocytes derived from tumor-bearing mice was enhanced by tumor-bearing lungs in a coculture system in vitro (Supplementary Fig. S1b). Because ZC3H12D was recognized as a tumor suppressor gene in follicular lymphoma and lung cancer patients[18,19], the role of ZC3H12D in metastasis was further examined. The primary tumor formed by E0771

implantation in Zc3h12d–/– mice[19,20] exhibited a similar growth rate to wild-type mice (Supplementary Fig. S1c), indicating that ZC3H12D in host cells may be an insufficient tumor suppressor of rapid primary tumor growth. However, lung metastasis was more severe in Zc3h12d–/– than in wild-type mice (Supplementary Fig. S1c), implying that ZC3H12D acted as a tumor suppressor molecule in metastasis.

To clarify the ZC3H12D localization pattern in immune cells stimulated by a tumor, ZC3H12D protein was observed in peripheral blood mononuclear cells (PBMCs) derived from mice without tumors and tumor-bearing mice using a flow cytometer. ZC3H12D was detected on the cell surface of PBMCs from mice without tumors (Fig. 1b, top). Note that the specificity of this ZC3H12D antibody was validated using a knockout sample. In contrast, in PBMCs from tumor-bearing mice, ZC3H12D was detected in the intracellular sphere but not on the cell surface (Fig. 1b, bottom). This tumor-mediated change in the ZC3H12D localization pattern was further confirmed using two other anti-ZC3H12D antibodies (Supplementary Fig. S1d). Tumor-conditioned medium (TCM), used as a tumor cell culture as it is expected to contain elements secreted from tumor cells, upregulated ZC3H12D 30 min after stimulation on splenic leukocytes (Fig. 1c). Three hours later, ZC3H12D protein completely moved inside the cell (Fig. 1c). Because ZC3H12D is considered an RNA-binding protein[21,22], it was suspected that nucleic acid was the explanatory factor in this phenomenon. Strikingly, pretreatment of TCM with RNase (TCM + RNase) did not affect the localization pattern of ZC3H12D (Fig. 1d, wt panel). The addition of RNA isolated from TCM resulted in the same phenomenon (Fig. 1d, wt panel, RNA-TCM column). These data suggested that RNA drives the translocation of ZC3H12D. In contrast, this was not observed when splenic leukocytes from Zc3h12d–/– mice were used (Fig. 1d).

**Nonvesicular interleukin-1β (IL1β)-mRNA was detected in lung TCM.** The tumor-related environment contains extracellular RNA (exRNA) associated with microvesicles, exosomes, and ribonucleoproteins (RNPs)[23,24]. Based on data that TCM induced the RNA-depending response of the ZC3H12D location (Fig. 1d), it was hypothesized that some specific exRNA[23] binds to ZC3H12D. Because exRNA should be exposed to ZC3H12D protein on leukocytes, the authors tried to examine which non-vesicular exRNA (nex-RNA) was increased in the lung microenvironment stimulated by a tumor. Microarray analysis was carried out on the lung EC TCM. The top 20 ranked genes are listed in Supplementary Table S2. The authors then searched for mRNA differences in expression between wild-type and Zc3h12d–/– mouse spleens. Spleens were chosen because they had the highest ZC3H12D expression compared to other organs tested in this study. The rationale was that ZC3H12D depletion effects might be seen by comparing wild-type and Zc3h12d–/– spleen data. This comparison revealed that immunoglobulin κ variable (Igkv), resistin like-γ (Retnlg), IL1β, and matrix metallopeptidase 8 (Mmp8) were upregulated in the Zc3h12d–/– sample (Supplementary Table S3). Among upregulated genes, this study focused on IL1β-mRNA because this gene exhibited relatively high expression of nex-RNA derived from TCM-stimulated lung cells (Supplementary Table S2). Also, βactin was chosen as a negative control in this study because it was abundantly found in TCM-stimulated lungs (Supplementary Table S2), and its expression levels did not change between wild-type and Zc3h12d–/– mice (Supplementary Table S3). First, this study tested whether IL1β-mRNA was detected in lung TCM prepared by the coculture of lung specimens with various TCMs using LLC, E0771, or B16 melanoma cells. To determine the relative

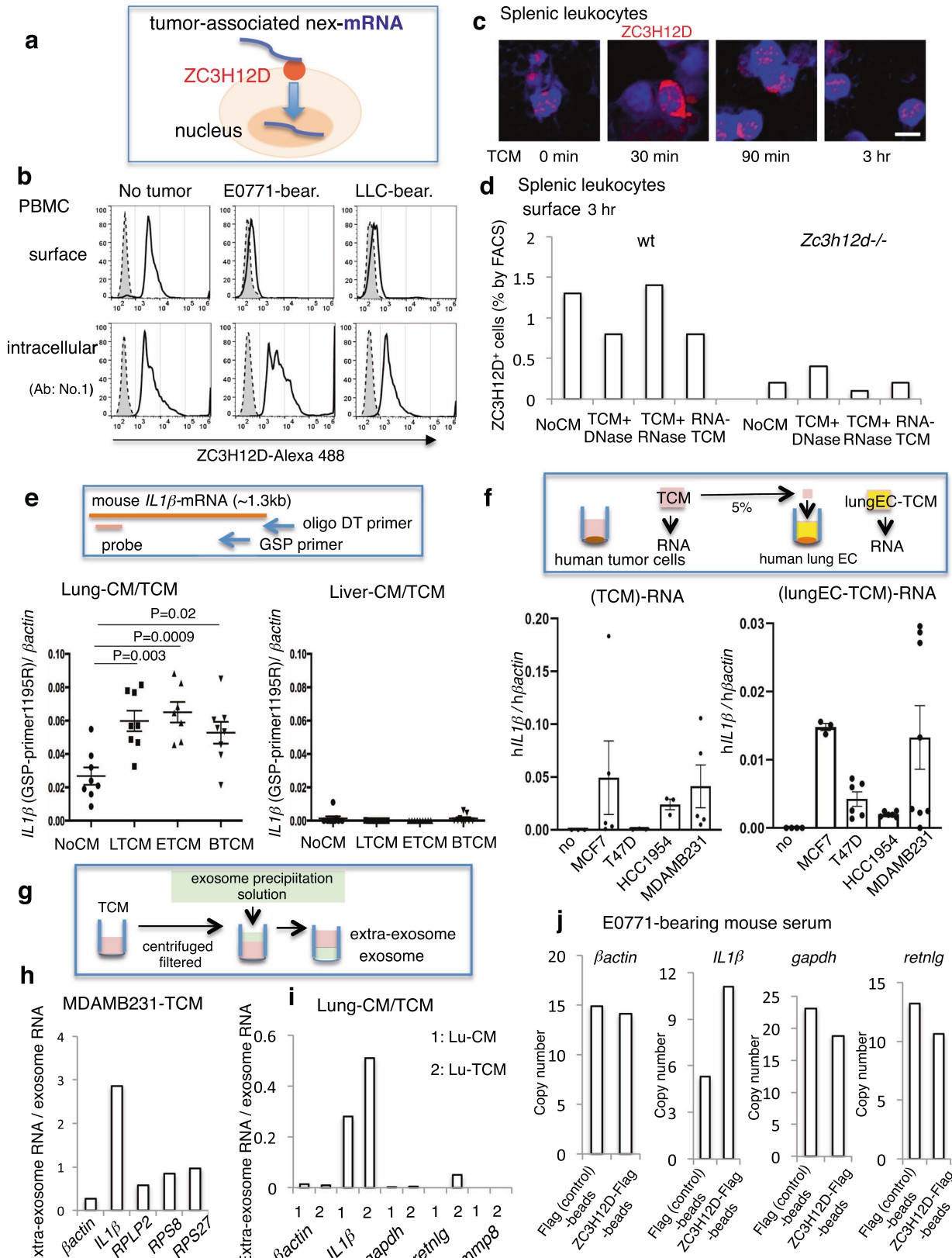

expression levels of *IL1β*-mRNA, a gene-specific primer (GSP) was used for the first-strand synthesis in quantitative reverse transcription-polymerase chain reaction (RT-qPCR; Fig. 1e, top). In Fig. 1e, lung TCMs contained larger amounts of *IL1β*-mRNA than nonstimulating lung CM, whereas liver TCMs had lower amounts than lung TCMs. To quantitate extracellular human (h)

*IL1β*-mRNA, CMs were collected from human breast cancer cells, including MCF7, MDAMB231, T47D, and HCC1954 (Fig. 1f, left). Along with these samples, h*IL1β*-mRNA was also analyzed in CM from human lung ECs stimulated by the aforementioned TCMs (Fig. 1f, right) because mouse lung culture medium stimulated with TCM contained mouse (m)*IL1β*-mRNA (Fig. 1e).

**Fig. 1 Detection of ZC3H12D on the cell surface and tumor-associated nex-*IL1β*-mRNA. a** Summary of the RNA uptake model in which ZC3H12D captures and transfers nex-mRNA to the nucleus. **b** FACS analysis of surface and intracellular ZC3H12D protein in PBMCs derived from mice without tumors or tumor-bearing mice. Several anti-ZC3H12D antibodies (bold line No. 1 in Fig. 1 and Nos. 2 and 3 in Supplementary Fig. S1d) and a control isotype antibody (dashed line) were used. **c** Kinetics of ZC3H12D expression in splenic leukocytes after stimulation by TCM. Bar, 5 μm. Experiments were repeated twice with similar results. Validation of anti-ZC3H12D antibody is shown in Supplementary Fig. S10 (validation and gating strategy). **d** Flow cytometric analysis of ZC3H12D on the surface of splenic leukocytes derived from wild-type and *Zc3h12d*–/– mice 3 h after stimulation of TCM treated with DNase or RNase. NoCM, control CM without tumor cells. The addition of RNase A to TCM did not cause intracellular entry of *ZC3H12D* (TCM + RNase). Reduction of surface ZC3H12D after application of purified RNA from TCM (RNA-TCM). ZC3H12D signals in *Zc3h12d*–/– mouse leukocytes were due to nonspecific antibody binding. **e** Detection system of *IL1β*-mRNA using two RT primers and a TaqMan qPCR probe (top). Real-time PCR analysis of *IL1β*-mRNA in lung CM cultured with various TCMs, such as LLC-TCM, E0771-TCM, and B16-TCM, presented as LTCM, ETCM, and BTCM, respectively (*n* = 8, 8, 7, and 8 biologically independent samples for NoCM, LTCM, ETCM, and BTCM, respectively). Faint detection of *IL1β*-mRNA in liver CM cultured with various TCMs (*n* = 9 biologically independent samples). **f** Scheme of RNA isolation from TCMs and CM of lung EC. qPCR detection of h*IL1β*-mRNA in TCMs after incubation of human breast cancer cells, including MCF7, MDAMB231, T47D, and HCC1954 (bottom left, *n* = 3, 5, 3, 3, and 5 biologically independent samples for no, MCF7, T47D, HCC1954, and MDAMB231, respectively). Secretion of h*IL1β*-mRNA from human lung ECs stimulated by various TCMs (bottom right, *n* = 4, 3, 6, 6, and 8 biologically independent samples for no, MCF7, T47D, HCC1954, and MDAMB231, respectively). "No," zero cancer cells. **g** Depiction of extraexosome and exosome RNA isolation. **h** Ratio of nonvesicular RNA (extraexosome RNA) compared to exosome RNA in MDAMB231-TCM. **i** Ratio of extraexosome RNA compared to exosome RNA in mouse lung culture medium with control CM or TCM. **j** qPCR analysis after direct trap of nonvesicular mRNA by ZC3H12D protein beads. Representative data are shown in the graph. Repeated experiment data related to **d** and **h**–**j** are shown in Supplementary Fig. S9. In the graphs, averages ± SEM and the results of Student's *t*-test or one-way ANOVAs are shown. The *P* values are shown in the figure. Source data are provided as a Source Data file.

Next, this study investigated whether there is nonvesicular *IL1β*-mRNA in human breast cancer TCM because it was reported that human glioma stem cell culture contained mRNA in EVs and nonvesicular RNPs[24]. The isolation conditions for nonvesicular (called extraexosome) and exosome fractions (Fig. 1g) we first set up, and the results were validated by RPLP2, RPS8, and RPS27 TaqMan probe-based qPCR (Fig. 1h). MDAMB231-TCM included more h*IL1β*-mRNA in the extraexosome fraction than in the exosome fraction (Fig. 1h). Extraexosomal m*IL1β*-mRNA in mouse lung TCM was increased (Fig. 1i) and considered to be PolyA(+), as they were reverse transcribed using oligo(dT) primer (Supplementary Fig. S1e). GSP-RT-qPCR also confirmed an increase of m*IL1β*-mRNA in the lung TCM (Supplementary Fig. S1e). Also, nonvesicular m*IL1β*-mRNA in TCM-stimulated mice was directly captured by ZC3H12D protein beads (Fig. 1j). In contrast, mRNA of *βactin*, *gapdh*, *retnlg*, *mmp8*, and *Igkv* were not enriched by bead pull-down (Fig. 1j; *mmp8* and *Igkv* were not detected).

**ZC3H12D binds to nex-*IL1β*-mRNA.** Whether nex-mRNA could bind to ZC3H12D on the cell surface was examined next. ZC3H12D was used because it is expressed in murine *RAW 264.7* macrophage cells[21,25] and human THP1 cells[26]. To visualize the interaction between *IL1β*-mRNA and ZC3H12D protein, an mZC3H12D-overexpressing RAW cell line (ZC⁺RAW) was set up, and fluorescein isothiocyanate (FITC)-labeled *IL1β*-mRNA was applied. Thirty minutes after TCM stimulation, the colocalization signals in ZC⁺RAW cells were more prominent (Pearson's *R* = 0.47 ± 0.04; *n* = 3) than cells stimulated by control NoCM (Fig. 2a, arrow and Supplementary Fig. S2a). In ZC3H12D⁺THP1 cells isolated using a cell sorter and an anti-ZC3H12D antibody, similar colocalization signals were observed after applying *IL1β*-mRNA-FITC (Supplementary Fig. S2b; Pearson's *R* = 0.62 ± 0.08; *n* = 3). Whether there was a direct interaction between *IL1β*-mRNA in lung TCM and ZC3H12D protein was also investigated. After application of lung TCM to ZC⁺RAW cells, cells were treated with ultraviolet (UV) irradiation to fix RNA on the proteins. Then, ZC3H12D protein with FLAG tag was isolated using anti-FLAG magnetic beads, and the amount of *IL1β*-mRNA bound to ZC3H12D-FLAG-beads was quantified by RT-qPCR. The data showed that ZC3H12D protein enriched *IL1β*-mRNA in ZC⁺RAW cells (Fig. 2b). In contrast, the

*gapdh*- and *βactin*-mRNA control showed little enrichment in the condition (Fig. 2b). Direct interactions between ZC3H12D protein and *IL1β*-mRNA were investigated using fluorescence correlation spectroscopy (FCS). This technique has been used to demonstrate a direct protein-RNA interaction in vitro[27]. The correlation curves of fluorescence-labeled ZC3H12D were generated in the presence of *βactin*- or *IL1β*-mRNA (Fig. 2c). The curve fitting revealed that FCS data are a mixture of three elements: ZC3H12D protein, ZC3H12D-RNA complex, and fast-decay component. The protein-RNA complex ratio was evaluated based on the ratio of the elements (Fig. 2d). FCS data indicate that the dissociation constant for ZC3H12D- *βactin*- and *IL1β*-mRNA were in the order of 1 nM. It was also confirmed that the 3′-UTR of *IL1β*-mRNA had direct interaction with ZC3H12D protein (Fig. 2e, f). Because the 3′-UTR of *IL1β*-mRNA had a different sequence from *βactin*-mRNA, the binding might include nonspecific and sequence-specific interactions in vitro.

**ZC3H12D transports nex-*IL1β*-mRNA into the nucleus.** To determine the intracellular kinetics of *IL1β*-mRNA captured by ZC3H12D, three-dimensional (3D) cell analyses were conducted using a series of z-stack confocal microscopy images. The 5′-Cap and 3′-Poly(A) are hallmark structures of eukaryotic mRNA[28,29]; therefore, the FITC-labeled naked *IL1β*-mRNA and its 5′-Cap and 3′-Poly(A) adduct were tested in this assay. Surprisingly, both were observed in the nucleus of ZC⁺RAW cells 3 h after application. *IL1β*-mRNA-Cap(+)PolyA(+)-FITC was more clearly observed than *IL1β*-mRNA-FITC in the nucleus, whereas the control *βactin*-mRNA-Cap(+)PolyA(+)-FITC gave almost no signal (Fig. 3a left). To evaluate whether exogenous *IL1β*-mRNA was translated to functionally modify the cell, *IL1β*-mRNA with nonsense mutations (*IL1β*-stop-mRNA) was prepared. This RNA has three stop codon mutations in the protein coding sequence (CDS) to not generate the IL1β protein. Interestingly, *IL1β*-stop-mRNA is clearly transported to the nucleus (Fig. 3a, right). ZC3H12D seemed to move into speckles based on the immunostaining data that ZC3H12D showed relatively dominant costaining with SC35, a speckle marker[30], rather than other nuclear body markers, such as promyelocytic leukemia (PML), fibrillarin, and SFPQ for PML body, nucleolus, and paraspeckle, respectively (Supplementary Fig. S2c, d). Also, the flow cytometric analysis of THP1 cells revealed that ZC3H12D cell surface

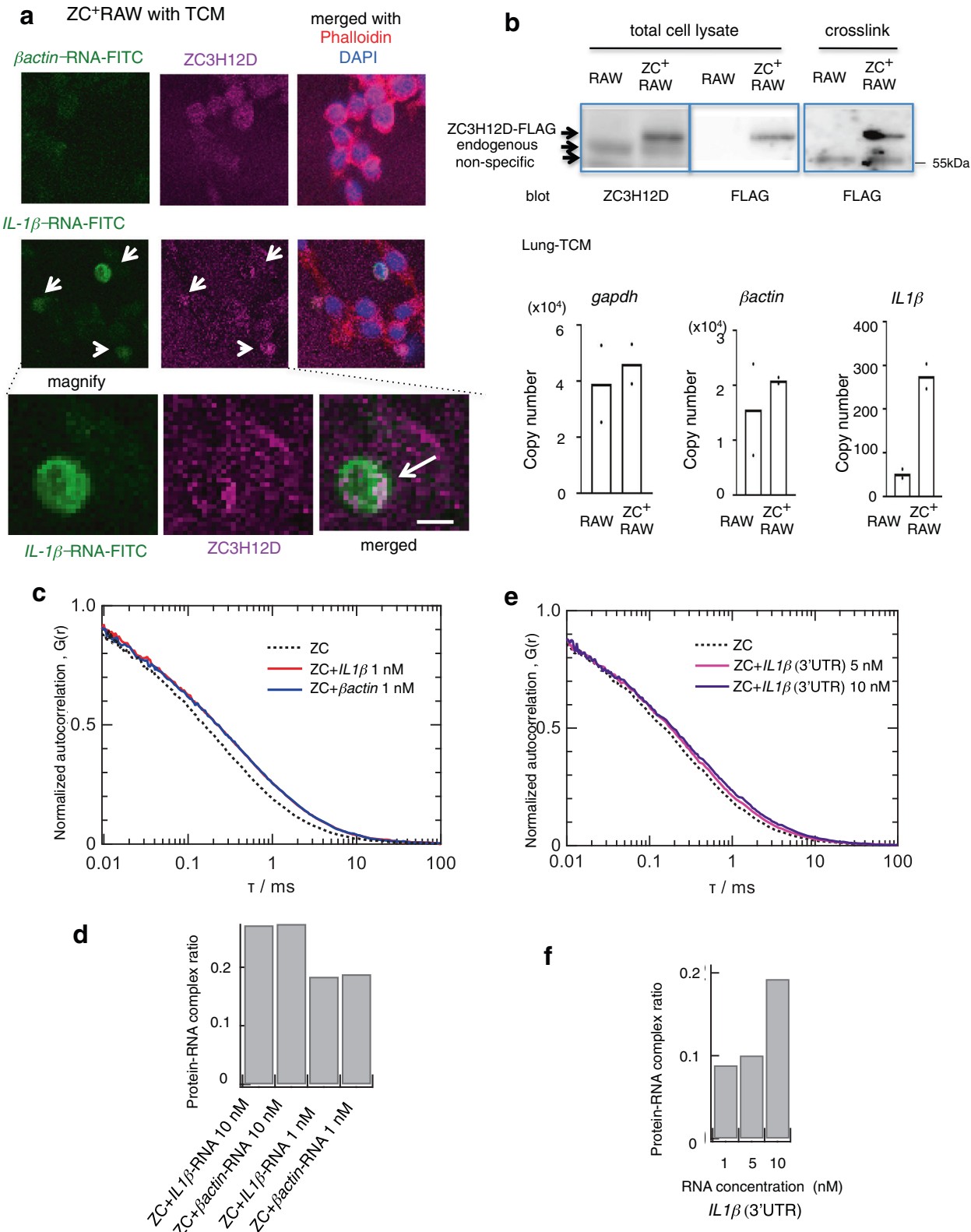

expression was observed in 0.3% of the cells with CM stimulation. In contrast, the population decreased to 0.1% in the presence of TCM stimulation. This decrease indicates that TCM stimulated the intracellular internalization of ZC3H12D from the cell surface. ZC3H12D+THP1 cells incorporated *IL1β*-mRNA and *IL1β*-stop-mRNA into the nucleus, although the signals were weaker than in NK cells (Supplementary Fig. S3a). To quantify the

amount of external RNA transported into the nucleus, a chimeric *IL1β*-mRNA was synthesized by combining 5′-h*IL1β*-mRNA and 3′-m*IL1β*-mRNA (Fig. 3b). Note that the qPCR probe was designed to amplify h*IL1β*-mRNA located in the 5′-region but not endogenous mouse transcripts (top, Fig. 3b; see Supplementary Fig. S3b for more detail). These data indicate that chimeric RNA was transported into the nucleus regardless of the Cap or Poly(A)

**Fig. 2 nex-*IL1β*-mRNA binds to ZC3H12D on the cell surface. a** Representative images showing the colocalization of *IL1β*-mRNA-FITC and ZC3H12D in ZC3H12D-overexpressing RAW cells (ZC⁺RAW) 30 min after TCM stimulation. The image after stimulation by NoCM as the control is shown in Supplementary Fig. S2a. Merged data with actin filament marker, phalloidin, and DAPI were shown. Bars, 5 μm. Experiments were repeated twice with similar results. **b** Western blot analysis of RAW and ZC⁺RAW. Anti-ZC3H12D and anti-FLAG antibodies detected FLAG-tagged (C-term) ZC3H12D, and immunoprecipitated ZC3H12D after TCM application plus UV-crosslink is shown (top). The binding of *IL1β*-mRNA in lung TCM (bottom) to ZC3H12D-FLAG was detected by qPCR analyses. The averages of two independent experiments were shown in the graphs. Repeated experiment data related to *IL1β*-RNA are shown in Supplementary Fig. S9. **c, e** Autocorrelation curves obtained by FCS measurements of the labeled ZC3H12D protein in the absence and presence of non-labeled RNA. Alexa Fluor 488-labeled ZC3H12D protein (1 nM, dotted line) was mixed with various concentrations of mouse *IL1β*-RNA (10 nM, solid (red) line) or *βactin*-RNA (10 nM, solid (blue) line) (**c**), and *IL1β*(3′-UTR)-RNA (5 nM, solid (magenta) line) or *IL1β*(3′-UTR)-RNA (10 nM, solid (purple) line) (**e**). **d, f** Fractions of the ZC3H12D-RNA complex relative to the total content of ZC3H12D obtained by a three-component analysis of the autocorrelation data. Source data are provided as a Source Data file.

modification (Fig. 3b, bottom left), although small amounts were detected in the cytoplasm (Fig. 3b, bottom right). To confirm the involvement of ZC3H12D in this phenomenon, this assay was repeated using B220⁺CD11c⁺NK1.1⁺ cells[16,31] derived from TCM-stimulated wild-type and *Zc3h12d–/–* mice. The localization of *IL1β*-mRNA-FITC and *IL1β*-stop-mRNA-FITC in the nucleus of wild-type mouse cells was prominent 3 h after the application (Fig. 3c). In contrast, there were minimal signals in the nuclei of those cells from *Zc3h12d–/–* mice (Fig. 3c).

**ZC3H12D recognizes the 3′-UTR of *IL1β*-mRNA**. A sequence-specific uptake via ZC3H12D protein was investigated using *IL1β*-mRNA. For this purpose, two partial *IL1β*-mRNA, a CDS and a 3′-UTR, were first prepared. The former consists of a short 5′-UTR and a CDS to give 897 nt; the latter is 451 nt long. These RNAs were tested in a cell uptake assay (Fig. 3d). The uptake of FITC-labeled *IL1β*-(3′-UTR)-mRNA in ZC⁺RAW was greater than that of *IL1β*-(CDS)-mRNA (Fig. 3d). Moreover, pre-incubation with *IL1β*-(3′-UTR)-mRNA blocked the uptake of FITC-labeled full-length *IL1β*-mRNA in the nucleus, but neither *IL1β*-(CDS)-mRNA nor *βactin*-mRNA had this effect (Supplementary Fig. S3c). Taken together, the 3′-UTR in *IL1β*-mRNA is recognized by ZC⁺RAW cells in a region-specific manner.

Some RNA-binding proteins recognize AU-rich elements (AREs) and/or stem-loop structures in 3′-UTR for controlling the stability of target mRNA[32]. In this study, an electrophoretic mobility shift assay (EMSA) revealed that ZC3H12D bound to specific sites of single-stranded RNA (ssRNA) with 50 nt in the 3′-UTR of *IL1β*-mRNA (Fig. 4a, No. 5 probe in Fig. 4b and Supplementary Fig. S4a). This binding was confirmed by inhibition assay using cold RNA oligonucleotide (Fig. 4a, c). These data indicate that ZC3H12D interacts with relatively large elements around AREs in *IL1β*-mRNA with no typical stem-loop sequence. Subsequent competition assays revealed that the binding site of ZC3H12D contains a typical AU-rich sequence (UAUUUAU) but that its short fragments (20 nt) exhibited much lower affinities than the 50 nt oligonucleotide (Fig. 4a, d and Supplementary Fig. S4b). In the nucleus, binding is expected to be lost under conditions with other ARE-binding proteins, such as HuR[33].

**RNA uptake may not be supported by Regnase-1**. Many ZF proteins are involved in intracellular RNA metabolism in innate immunity[32]. Among the ZC3H12 family, ZC3H12A (also known as Regnase-1) regulates *IL6*-mRNA with binding to the stem-loop in the 3′-UTR[34–36]. Also, ZC3H12D potentially interacts with ZC3H12A[21]. Whether ZC3H12A had a cooperative function in the ZC3H12D-dependent uptake of *IL1β*-mRNA was therefore investigated. In the microarray data (Supplementary Table S3), *Zc3h12a* expression levels in the *Zc3h12d–/–* spleen were close to those in the wild-type spleen. The gene expression profiles of

B220⁺CD11c⁺NK1.1⁺NK cells in tumor-bearing lungs and livers were also examined. The data showed that *Zc3h12d* was markedly upregulated during the relocation of cells from the liver to the lungs, whereas *Zc3h12a* expression was constant (Supplementary Table S1); this suggested that there is no direct correlation between the expression of *Zc3h12a* and *Zc3h12d*. The signal of ZC3H12D revealed a similar pattern in B220⁺CD11c⁺NK1.1⁺ NK cells derived from *Zc3h12a⁻/⁻* (*Regnase-1–/–*) mice and wild-type mice (Supplementary Fig. S5a). Concordantly, NK cells derived from both genotypes exhibited *IL1β*-mRNA uptake in the nucleus (Fig. 5a, right; Supplementary Fig. S5b) when those cells derived from *Zc3h12d–/–* mice lost the uptake again (Fig. 5a, left; Supplementary Fig. S5b). These data indicated that ZC3H12A might not be involved in the ZC3H12D-dependent uptake system in B220⁺CD11c⁺NK1.1⁺NK cells.

**Entrained *IL1β*-mRNA in the nucleus may have functions**. When *IL1β*-mRNA with or without Cap(+)PolyA(+) was applied to B220⁺CD11c⁺NK1.1⁺ cells, *IL1β*-mRNA-Cap(+) PolyA(+)-FITC was more clearly detected than *IL1β*-mRNA-FITC in the nucleus of B220⁺CD11c⁺NK1.1⁺ cells from wild-type mice. In contrast, *IL1β*-mRNA-FITC was pooled outside of the nuclei of *Zc3h12d–/–* mice. Interestingly, some wild-type cells showed an irregular nucleus 6 h after application of 20–100 ng/mL Cap(+)PolyA(+) (Supplementary Fig. S5c, 100 ng/mL). To remove the possibility that this irregular nucleus shape was due to the addition of FITC-modified substances, the same phenomenon was observed using unlabeled *IL1β*-mRNA. In this case, an increase in histone H2AX phosphorylation, a marker for DNA stress/damage[37], was observed in the nucleus of B220⁺ CD11c⁺NK1.1⁺ cells from wild-type mice but not observed in *Zc3h12d–/–* mice (Fig. 5b). B220⁺CD11c⁺NK1.1⁺ cells derived from *Regnase-1–/–* mouse exhibited H2AX phosphorylation upon incorporation with *IL1β*-mRNA- Cap(+)PolyA(+) (Supplementary Fig. S5d). Note that H2AX performs both structural and functional roles in chromatin regulation beyond a specific DNA double-strand break marker[37]. Next, this study tried to determine whether entrainment of *IL1β*-mRNA in the nucleus induces cell death because of the increase of cellular stress (Fig. 5c). The time course of cell death after treatment with *βactin*-mRNA, *IL1β*-mRNA and *IL1β*-stop-mRNA was assessed. In this assay, B220⁺ CD11c⁺NK1.1⁺NK cells derived from TCM-stimulated wild-type and *Zc3h12d–/–* mice showed a slight increase of necrosis signals (Fig. 5c). When 20 ng/mL *IL1β*-mRNA and *IL1β*-stop-mRNA were applied to B220⁺CD11c⁺NK1.1⁺NK cells from wild-type, signal intensities were suppressed compared to those with *βactin*-mRNA (Fig. 5c). B220⁺CD11c⁺NK1.1⁺NK cells from *Zc3h12d–/–* mice showed almost no response for any RNA (Fig. 5c). This indicates that the incorporation of *IL1β*-mRNA in the nucleus may contribute to NK cell survival.

Whether incorporation of *IL1β*-mRNA into the nucleus affects intranuclear activities, such as transcription, was then examined.

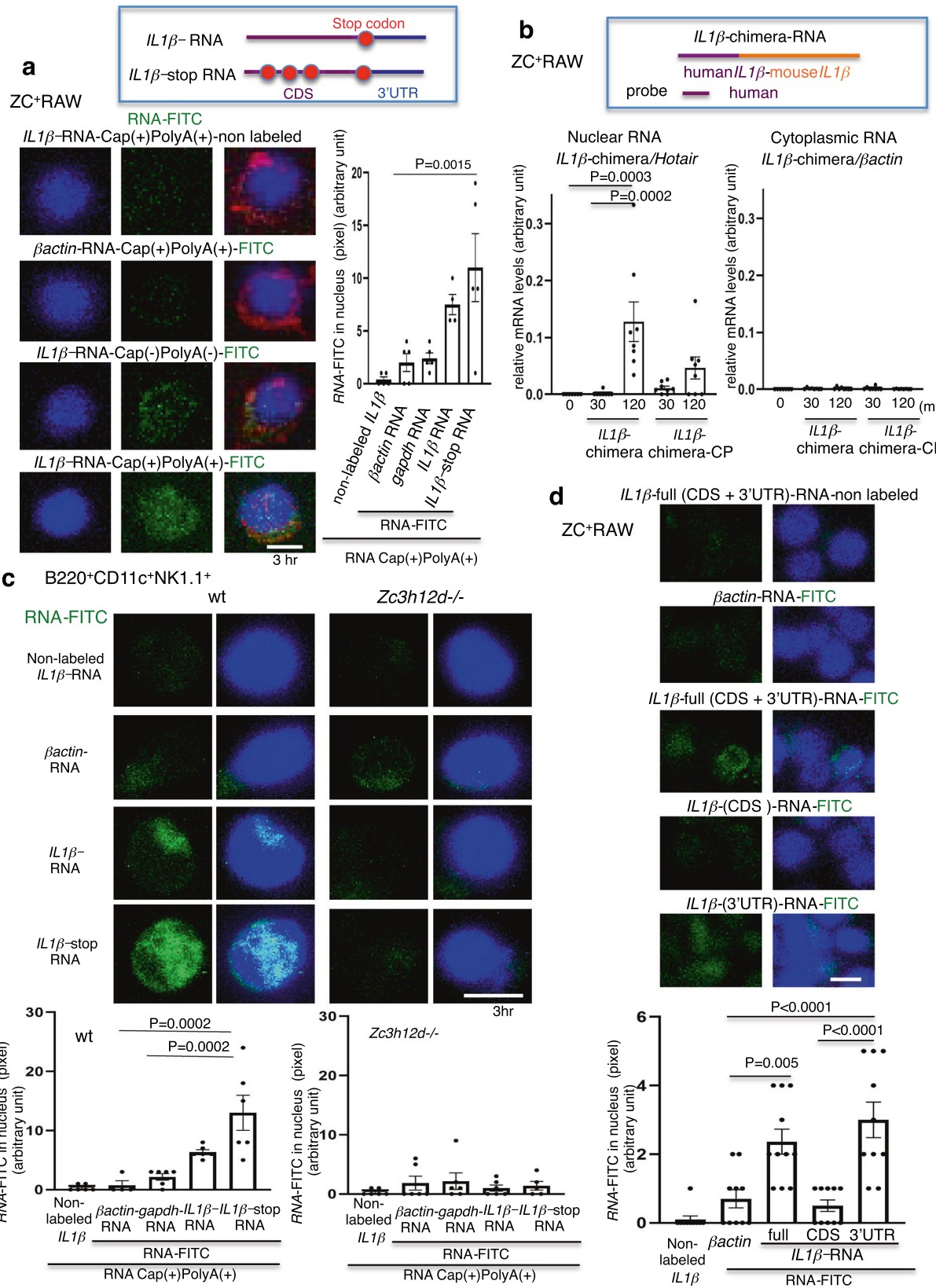

This study first examined whether nuclear-transported *IL1β*-mRNA was associated with RNA polymerase II (Pol II), regulating the initiation and elongation of transcription of mRNA[38]. To distinguish the applied *IL1β*-mRNA from endogenous *IL1β*-mRNA, a qPCR probe that anneals human-specific sequences only was designed. After the application of chimera *IL1β*-mRNA to ZC⁺RAW, the transported chimera *IL1β*-mRNA was fixed with UV irradiation and coimmunoprecipitated by an anti-Pol II antibody (Fig. 5d). This study then searched for the gene expression changes affected by *IL1β*-mRNA uptake in the same microarray data (Supplementary Table S3). Seven genes were picked based on the following criteria that would suggest

**Fig. 3 ZC3H12D brings nex-*IL1β*-mRNA into the nucleus. a** Fluorescence confocal microscopy of ZC⁺RAW after application of 10 ng/mL non-labeled *IL1β*-mRNA, FITC-labeled *βactin*-mRNA, and *IL1β*-mRNA with or without Cap-Poly(A) (CP). 3D images are shown (left). Quantitative analysis of *RNA*-FITC in the nucleus (right). *IL1β*-stop-mRNA has three stop codon mutations in the CDS (top). The non-labeled *IL1β*-mRNA was used as a basal control for the FITC signal (n = 5, 5, 5, 4, and 5 cells for non-labeled *IL1β*-RNA, FITC-labeled *βactin*-RNA, *gapdh*-RNA, *IL1β*-RNA, and *IL1β*-stop-RNA respectively). Repeated experiment data are shown in Supplementary Fig. S9. **b** System for the detection of external *IL1β*-mRNA using human-mouse chimera *IL1β*-mRNA (*IL1β*-chimera) followed by reverse transcription with a 3′-mGSP and a 5′-human probe in the RT-qPCR analysis (top; more details in Supplementary Fig. S3b). Detection of *IL1β*-chimeric-RNA entry into ZC⁺RAW. Nuclear RNA and cytoplasmic *Hotair* and *βactin* normalized nuclear RNA and cytoplasmic RNA, respectively (n = 7, 8, 8, 8, and 8 biologically independent samples for 0, 30, and 120 min with *IL1β*-chimera, 30 and 120 min with *IL1β*-chimera-CP, respectively). **c** 3D images showing *IL1β*-mRNA-FITC and *IL1β*-stop-mRNA-FITC uptake in the nucleus of B220⁺CD11c⁺NK1.1⁺ cells derived from TCM-stimulated wild-type and *Zc3h12d−/−* mice 3 h after the application of 10 ng/mL RNA (top). Quantitative analysis of *RNA*-FITC in the nucleus of B220⁺CD11c⁺NK1.1⁺ cells from wild-type mice (bottom left, n = 6, 4, 6, 6, and 6 cells for non-labeled *IL1β*-RNA, FITC-labeled *βactin*-RNA, *gapdh*-RNA, *IL1β*-RNA, and *IL1β*-stop-RNA respectively) and from *Zc3h12d−/−* mice (bottom right, n = 6, 6, 6, 6, and 5 cells for non-labeled *IL1β*-RNA, FITC-labeled *βactin*-RNA, *gapdh*-RNA, *IL1β*-RNA, and *IL1β*-stop-RNA respectively). **d** Fluorescence confocal microscopy of ZC⁺RAW after the addition of FITC-labeled *βactin*-mRNA, full length, CDS, or 3′-UTR of *IL1β*-mRNA (top). Quantitative analysis of 10 ng/mL *IL1β*-mRNA domain-dependent nucleus uptake (bottom; n = 10, 10, 11, 10, and 10 cells for non-labeled *IL1β*-RNA, FITC-labeled *βactin*-RNA, full length, CDS, and 3′UTR of *IL1β*-RNA, respectively). In this analysis, 15-stacked 3D images were obtained from the bottom to the top of the nucleus. Bar, 5 μm. In the graphs, averages ± SEM and the results of one-way ANOVA with Bonferroni correction are shown. The *P* values are shown in the figure. Source data are provided as a Source Data file.

that their behavior is similar to that of *IL1β*: they were classified as nucleus-related by Gene Ontology and showed a relatively higher expression level in *Zc3h12d−/−* than in wild-type mice in the microarray data set (Supplementary Table S4). The gene expression levels of these seven genes were compared between RAW and ZC⁺RAW cells after treatment with *IL1β*-mRNA. qPCR revealed that *Dusp1*[39,40] and *IL1rn* were expressed at higher levels in ZC⁺RAW than in control RAW (Supplementary Fig. S6), and the content of these transcripts in nuclear RNA increased in conjunction with the uptake of *IL1β*-mRNA (Fig. 5e). Furthermore, *Dusp1* and *IL1rn* expression was upregulated in B220⁺CD11c⁺NK1.1⁺NK cells after application of *IL1β*-mRNA (Fig. 5f).

**ZC3H12D entrapment of *IL1β*-mRNA induces antimetastatic activity.** Whether *IL1β*-mRNA elicits migration activity in ZC3H12D⁺NK cells because those cells migrated from the liver to the lungs in tumor-bearing mice was investigated. B220⁺CD11c⁺NK1.1⁺NK cells derived from wild-type mice exhibited migration activity with *IL1β*-mRNA, but not *βactin*-mRNA, and this migration activity was absent in *Zc3h12d−/−* mouse cells (Fig. 6a). To demonstrate that ZC3H12D protein can capture naturally produced *IL1β*-mRNA, migration assays were conducted using nex-RNA isolated from TCM. To confirm the effect of ZC3H12D protein, migration assays were repeated using TCM passed through ZC3H12D-FLAG tag bound on anti-FLAG-beads. In this experiment, TCM passed through ZC3H12D protein-bound beads [designated as ZC3H12D(+) column] and passed through anti-FLAG beads [designated as ZC3H12D(−) column, used as a control] were prepared. nex-RNA were isolated from both TCMs and used for the migration assay. This migration assay revealed that nex-RNA treated with the ZC3H12D(+) column had reduced chemotactic activity compared to that of nex-RNA treated with the ZC3H12D(−) column in B220⁺CD11c⁺NK1.1⁺NK cells (Supplementary Fig. S7a), suggesting that *IL1β*-mRNA-ZC3H12D interactions play a vital role in cell migration activity.

This study tried to evaluate the effect of *IL1β*-mRNA on IFN-γ production in B220⁺CD11c⁺NK1.1⁺NK cells. Cells derived from wild-type mice were clearly stimulated by *IL1β*-mRNA, but only a faint induction was observed in cells from *Zc3h12d−/−* mice (Fig. 6b). Furthermore, to determine the tumoricidal effects of B220⁺CD11c⁺NK1.1⁺NK cells, rhodamine-labeled E0771 cells were cocultured with *IL1β*-mRNA-primed NK cells to count dead tumor cells stained with Zombie, a green fluorescent dye. In this assay system, to account for the effects of RNA priming on the tumor cells, NK cells were washed before being applied to tumor cells. These data revealed that *IL1β*-mRNA priming increased the tumoricidal activity of NK cells originating from wild-type mice but not from *Zc3h12d−/−* mice (Fig. 6c). Again, B220⁺CD11c⁺NK1.1⁺NK cells from *Zc3h12d+/−* mice had similar tumoricidal activity (Supplementary Fig. S7b). To check if *IL1β*-mRNA-primed macrophages enhance this activity, NK cells were coincubated with bone marrow-derived macrophages (BMDMs) stimulated by *IL1β*-mRNA (Fig. 6d, top). Both *IL1β*-mRNA-primed BMDMs derived from wild-type and *Zc3h12d−/−* mice did not have additive effects on tumoricidal activity in vitro (Fig. 6d, lower).

As ZC3H12D recognized the 3′-UTR of *IL1β*-mRNA, the functional region in the 3′-UTR was further examined. To define the functional element in the 3′-UTR of *IL1β*-mRNA, it was further divided into four fragments containing 150 nt, where each fragment had at least one 50 nt overlap with its neighboring fragments. The third fragment showed higher activity than the others in the migration assay of B220⁺CD11c⁺NK1.1⁺NK cells derived from wild-type mice but remained unnoticed from *Zc3h12d−/−* mice (Supplementary Fig. S7c). Next, NK activation via ZC3H12A[32,34,35], a subfamily of ZC3H12 proteins, was examined. In a migration assay, full-length and 3′-UTR *IL1β*-mRNA induced chemotactic activity in NK cells from wild-type mice and *Zc3h12a−/−* (*Regnase-1−/−*) mice[34], whereas CDS *IL1β*-mRNA produced almost the same activity levels as those from control RNA (Supplementary Fig. S7d). In addition, IFN-γ induction by *IL1β*-mRNA was similar in cells derived from both genotypes (Supplementary Fig. S5e). Thus, these data indicated that a central region of the 3′-UTR in *IL1β*-mRNA is recognized in a ZC3H12D protein-dependent manner.

In cancer, IL1β protein is prevalent in tumor promotion in association with proinflammatory immune cells[41]. Whether *IL1β*-mRNA-primed B220⁺CD11c⁺NK1.1⁺NK cells have an antimetastatic ability in vivo was investigated. In this assay system, NK cells were obtained from TCM-stimulated mouse spleens and primed with synthetic *IL1β*-mRNA. These NK cells were then applied to another TCM-stimulated mouse before injection of rhodamine-labeled tumor cells via the tail vein. To evaluate lung metastasis, lungs were isolated after injection, and the cell numbers in the lungs were counted (Fig. 6e, model). *IL1β*-mRNA-primed wild-type B220⁺CD11c⁺NK1.1⁺ cells reduced the number of metastatic tumor cells homing into the lungs (Fig. 6f). In contrast, *IL1β*-mRNA-primed NK cells from *Zc3h12d−/−* mice have only a slight antitumor effect (Fig. 6f, graph).

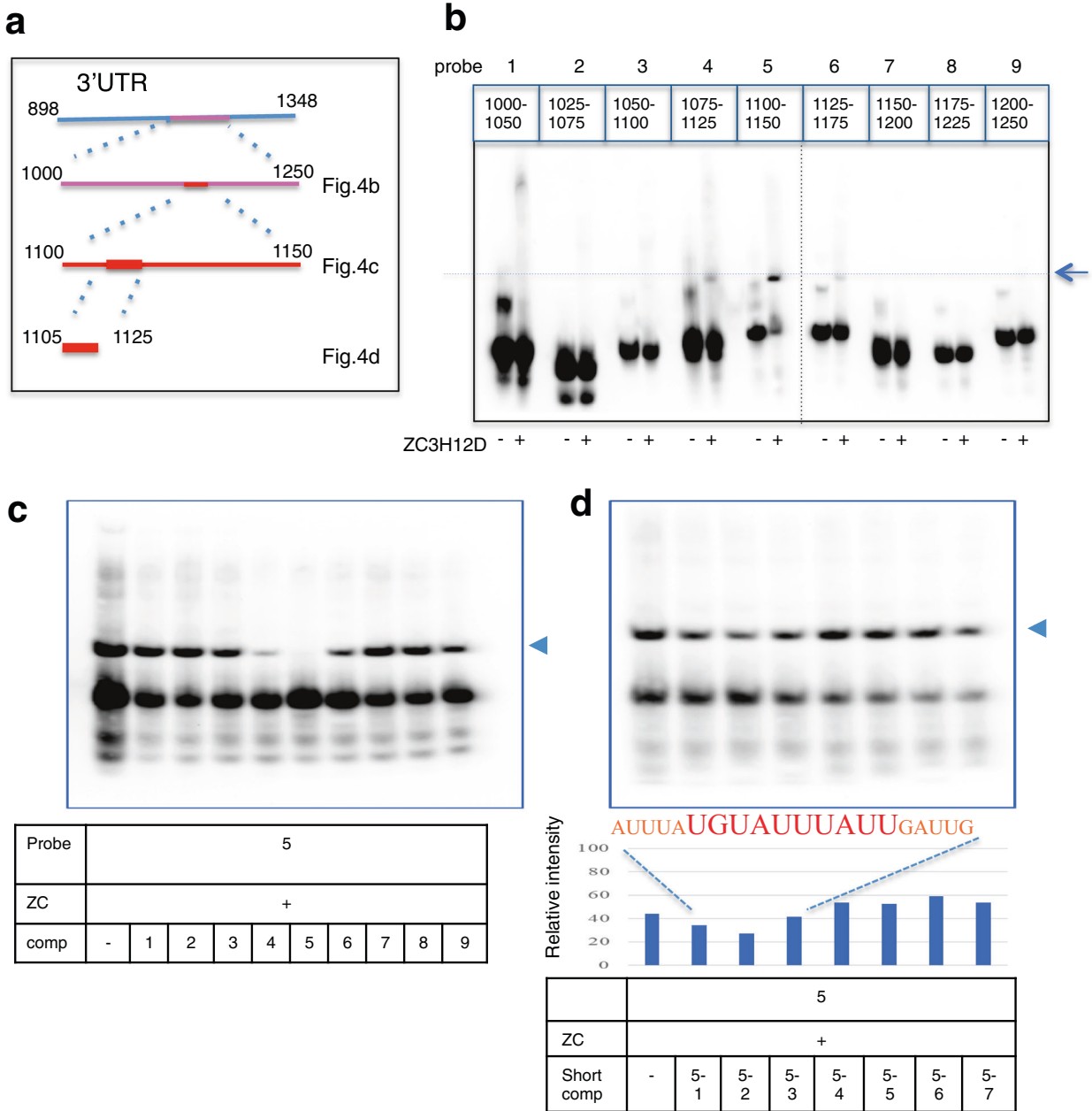

**Fig. 4 ZC3H12D recognizes the specific sequence of *IL1β*-mRNA. a** Scheme to determine ZC3H12D binding site in m*IL1β*-mRNA (NM_008361) 3′-UTR. Each number denotes nucleotide position. **b** EMSA. m*IL1β* probes (EMSA probes 1–9; see Supplementary Fig. S4a for sequences. Nucleotide positions of each probe are displayed at the top) were assayed with or without ZC3H12D protein. Probes 1–9 are 50 nt long, and the 3′-end is biotin-labeled. Arrow indicates the band shift due to the binding of hZC3H12D on probe 5. **c** EMSA competition assay. ZC3H12D protein and biotin-labeled probe 5 (50 nM) were mixed with a 100-fold excess amount of non-labeled probes 1–9 (5 μM). **d** EMSA competition assay. ZC3H12D protein and biotin-labeled probe 5 (50 nM) were mixed with a 100-fold excess amount of the non-labeled probes 5-1–5-7 (5 μM). The competitors are 20 nt long and part of probe 5 (Supplementary Fig. S4b). The graph indicates the relative intensity of the top band [= top band / (top band + bottom band) × 100]. The top band position is marked with an arrow. Experiments were repeated twice with similar results. Source data are provided as a Source Data file.

**Human NK cell has ZC3H12D-*IL1β*-mRNA axis.** Whether hZC3H12D plays a significant role in the mRNA uptake system was investigated. hZC3H12D has two alternatively spliced gene products[26]: one is a long form (58 kDa: hZC58), the primary structure of which is very similar to that of mouse protein, and the other is a short form (36 kDa: hZC36), which lacks a C-terminal domain. These two isoforms share the RNA-binding and ZF domains, but the proline-rich domain, which is considered an interaction site for ZC3H12A protein, is unique to hZC58 (Fig. 7a, top). For a mRNA uptake assay, 786-O cells, human renal carcinoma cells stably expressing hZC58 or hZC36, were first

established. These IHC analyses revealed that ZC3H12D was widely distributed in the cell membrane and cytoplasm of hZC58 cells, whereas it was found in a relatively limited area in hZC36 cells (Fig. 7a, bottom). These cells were incubated with FITC-labeled h*IL1β* (h*IL1β*-mRNA), and the resulting images indicated that h*IL1β*-mRNA-FITC was detected more often in the nucleus of hZC58 cells than in hZC36 cells (Fig. 7b), indicating that the C-terminal domain of hZC58 containing the proline-rich domain plays a vital role in the transportation of mRNA into the nucleus. The uptake of h*IL1β*-mRNA in human CD56+CD3-NK[42,43] cells isolated from human PBMCs (hPBMCs) was next examined.

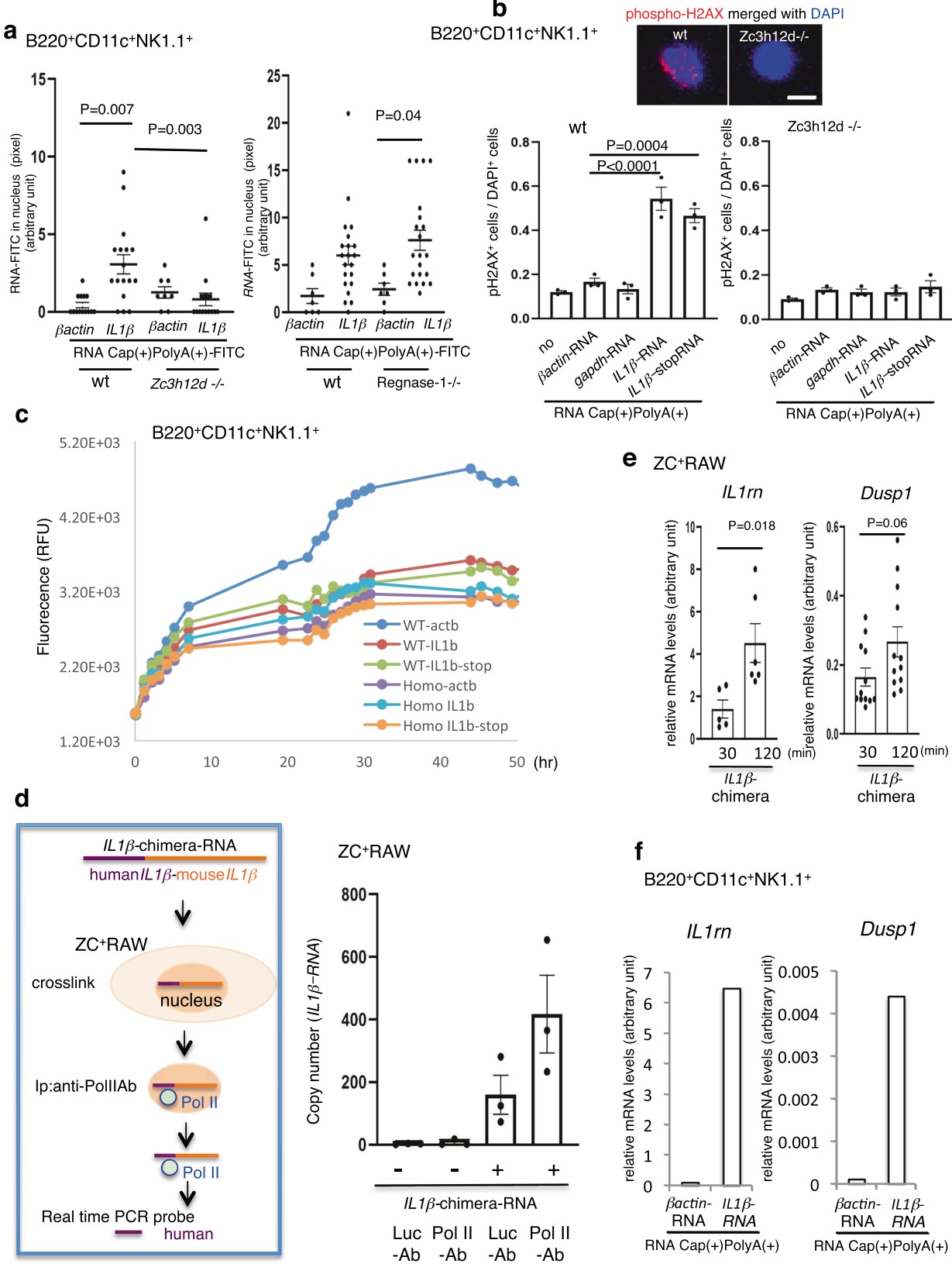

In flow cytometric analysis, ZC3H12D was detected on the cell surface of 46% of CD56$^+$CD3$^-$NK cells. FITC-labeled h$IL1\beta$-mRNA, but not control h$\beta actin$-mRNA, was incorporated into the nucleus of CD56$^+$CD3$^-$NK cells (Fig. 7c). Small interfering RNA (siRNA)-mediated knockdown of ZC3H12D in CD56$^+$CD3$^-$NK cells was carried out to elucidate whether ZC3H12D protein was involved in the NK cell uptake process. The suppression of ZC3H12D protein levels by siRNA electroporation (Supplementary Fig. S8a) clearly reduced h$IL1\beta$-mRNA uptake (Fig. 7d) in comparison to control siRNA- or $ZC3H12A$-siRNA-transfected cells.

**Fig. 5 nex-*IL1β*-mRNA induces biology in the ZC3H12D⁺NK nucleus. a** RNA uptake in B220⁺CD11c⁺NK1.1⁺ cells derived from wild-type, *Zc3h12d–/–*, and *Regnase-1–/–* (*Zc3h12a⁻/⁻*) mice 3 h after the application of 10 ng/mL *βactin*-mRNA-CP-FITC and *IL1β*-mRNA-CP-FITC. The quantifications of the FITC signal in B220⁺CD11c⁺NK1.1⁺ cell nucleus are shown (left, n = 14, 17, 8, and 15 cells from wild-type and *Zc3h12d–/–* mice for *βactin*-RNA and *IL1β*-RNA, respectively. right, n = 7, 21, 7, and 21 cells from wild-type and *Regnase-1–/–* mice for *βactin*-RNA and *IL1β*-RNA, respectively; 15 z-stack images per cell). Ages of *Regnase-1–/–* mice were close to *Zc3h12d–/–* mice. **b** Representative immunostaining for phospho-H2AX in the nucleus of B220⁺CD11c⁺NK1.1⁺ cells 3 h after the uptake of *IL1β*-mRNA-CP (top). Quantitative analysis of phospho-H2AX in B220⁺CD11c⁺NK1.1⁺ cells derived from wild-type and *Zc3h12d–/–* mice 3 h after the uptake of *βactin*-mRNA-CP *gapdh*-mRNA-CP *IL1β*-mRNA-CP and *IL1β*-stop-mRNA-CP (n = 3 biologically independent samples). Bar, 5 µm. **c** Time course of secondary necrosis in B220⁺CD11c⁺NK1.1⁺ cells derived from TCM-stimulated wild-type and *Zc3h12d–/–* (Homo) mice in the application of 20 ng/mL RNA. *βactin*-mRNA-CP, *IL1β*-mRNA-CP, and *IL1β*-stop-mRNA-CP were depicted as actb, IL1b, and IL1b-stop, respectively. Real-time tracking of fluorescein-DNA dye indicates the loss of membrane integrity resulting in late-stage apoptosis. **d** Detection system of entrained ex-*IL1β*-mRNA bound to Pol II in ZC⁺RAW cells (left). Real-time PCR analysis of *IL1β*-chimera-mRNA associated with Pol II in the nucleus. Anti-luciferase (Luc) antibody is the control (n = 3 biologically independent samples). **e** qPCR-derived quantifications of *IL1rn* and *Dusp1* expression in nuclear RNA of ZC⁺RAW after application of *IL1β*-chimera-mRNA (n = 5 and 6 biologically independent samples at 30 and 120 min, and n = 12 biologically independent samples at 30 and 120 min for IL1rn and Dusp1, respectively). **f** qPCR-derived quantifications of *IL1rn* and *Dusp1* expression in B220⁺CD11c⁺NK1.1⁺ cells after the application of *βactin*-mRNA and *IL1β*-mRNA. Representative data are shown in the graph. Repeated experiment data are shown in Supplementary Fig. S9. In the graphs, averages ± SEM and the results of Student's t-test (two-sided) or one-way ANOVA with Bonferroni correction are shown. The P values are shown in the figure. Source data are provided as a Source Data file.

NK activation after uptake of h*IL1β*-mRNA was examined. IFN-γ immunostaining revealed that CD56⁺CD3⁻NK cells increased in IFN-γ positivity after h*IL1β*-mRNA stimulation (Fig. 7e), and their IFN-γ expression levels were as high as those induced by recombinant IL12[44] as an IFN-γ inducer for NK cells[45]. Human NK cells are divided into two distinct subsets, CD56^bright and CD56^dim[46], and the tumor-related microenvironment is differently affected in those NK cells on their tumoricidal and proliferative functions[47]. As ZC3H12D was expressed in both subsets (Supplementary Fig. S8b), CD56^bright and CD56^dim NK cell fractions from hPBMCs were purified, and h*IL1β*-mRNA uptake was tested separately to find that it was in 38% of CD56^bright and 50% of CD56^dim NK cells. To mimic the tumor microenvironment, NK cells were first stimulated with TCM derived from human breast cancer MDAMB231 cells, and IFN-γ production response was then observed upon application of IL-2 protein and/or h*IL1β*-mRNA (Fig. 7f, top). Three days after TCM stimulation, h*IL1β*-mRNA but not IL2 protein increased IFN-γ production in CD56^dim NK cells (Fig. 7f, bottom left). The second stimulation by h*IL1β*-mRNA induced IFN-γ production at the same level as the first stimulation (Fig. 7f, bottom right; Supplementary Fig. S8c). Conversely, CD56^bright NK cells did not show induction (Supplementary Fig. S8d). The NK cell activation marker, NKG2D, in CD56^bright NK cells was slightly increased after the application of h*IL1β*-mRNA (Supplementary Fig. S8e).

In summary, ZC3H12D in NK cells selectively captures nex-*IL1β*-mRNA enriched in the tumor-stimulated lung microenvironment and transports it into the nucleus. To localize ZC3H12D protein at the cell surface, its proline-rich C-terminal domain is necessary, although it is apart from the RNA-binding site. *IL1β*-mRNA transport into the nucleus elicited three functions: (1) enhancing tumoricidal activity with cell migration ability and IFN-γ production, (2) regulation of antiapoptotic gene expression by associating with RNA Pol II, and (3) controlling the NK cell survival and prolonged IFN-γ production in a tumor microenvironment (Fig. 8).

## Discussion

Human serum and plasma contain various classes of RNA molecules, such as mRNA, microRNA, Piwi-interacting RNA, transfer RNA, and other miscellaneous ncRNA molecules[48–51]. Recently, nonvesicular RNP complexes, including UTR and the CDS of mRNA, were reported in human glioma cultured medium, indicating the existence of long types of mRNA[24]. This revealed that primary tumor-stimulated lung but not liver contained nonvesicular full-length *IL1β*-mRNA. Human full-length *IL1β*-mRNA with Poly(A) was more detected as an unpacked

vesicular form than packing form in EVs in breast cancer-derived TCM. Furthermore, the specific interaction of *IL1β*-mRNA was mediated by ZC3H12D protein, which recognizes its 3′-UTR; this is not surprising, as many mRNA-binding proteins bind to secondary structures, including stem loops and short consensus sequences, such as AREs, which are situated in the 3′-UTR of many transcripts in the cytoplasm[32]. Also, extracellular double-stranded RNA is nonspecifically recruited into the cytoplasm by the S1D family, resulting in the usage of cellular RNA interference silencing[52–54]. So far, several types of RNA uptake inside cells have been reported, and some of them are utilized in gene therapies[55]. Most importantly, this study revealed that ZC3H12D captures nex-mRNA at the cell surface; then, the mRNA-bound complex is transported into the nucleus across the cytoplasm. ZC3H12D belongs to the ZC3H12 family and shares an N-terminal domain (NTD), PilT N-terminus-like (PIN) domain, and ZF domain with other members. Among them, ZC3H12A (Regnase-1) is the most deeply investigated. Its biochemical studies revealed that recombinant Reganse-1 (NTD-PIN-ZF) recognized the stem-loop element in the 3′-UTR of *IL-6* mRNA[36] and degraded it in an Mg²⁺-dependent manner[56]. Reganse-1 and ZC3H12D regulate mRNA decay by recognizing the 3′-UTR of *IL-2*, *IL-6*, *IL-10*, and *TNFα*[20]. There is no surprise that both proteins regulated the same RNA. Regarding the PIN-ZF domain sequence, Regnase-1 and ZC3H12D are similar to each other, and their aspartate residues requiring a cleavage reaction are entirely conserved. In contrast, *IL-17a* mRNA degradation was regulated by ZC3H12D but not Regnase-1[20]. This minor difference in the enzymatic specificity is attributed to the difference in their amino acid sequences. The homology between these two proteins in the NTD is relatively low (45%). Thus, it is assumed that the NTD modified the biochemical functions of ZC3H12D. It is also assumed that ARE containing RNA bound to ZC3H12D was apart from Mg²⁺ sitting at the catalytic center so that it was not degraded as other stem-loop substrates. Furthermore, ZC3H12D binding to long synthetic RNA with ARE (50 nt) is much stronger than short synthetic RNA (20 nt), implying that the binding affinity of ZC3H12D is susceptible to various structural factors. The FCS measurements unveiled that ZC3H12D has a non-specific binding site for long (>1000 nt) RNA. Taken together, it is speculated that long RNA with ARE triggers a structural change in the ZC3H12D-RNA complex to give it biological functions. Phylogenetic relationship analysis revealed that the ZC3H12 family ancestor, nematode protein REGE-1, targets *ETS-4*-mRNA different from target genes by mammalian ZC3H12 and controls proinflammatory mRNA[57]. It is suggested that a unique substrate

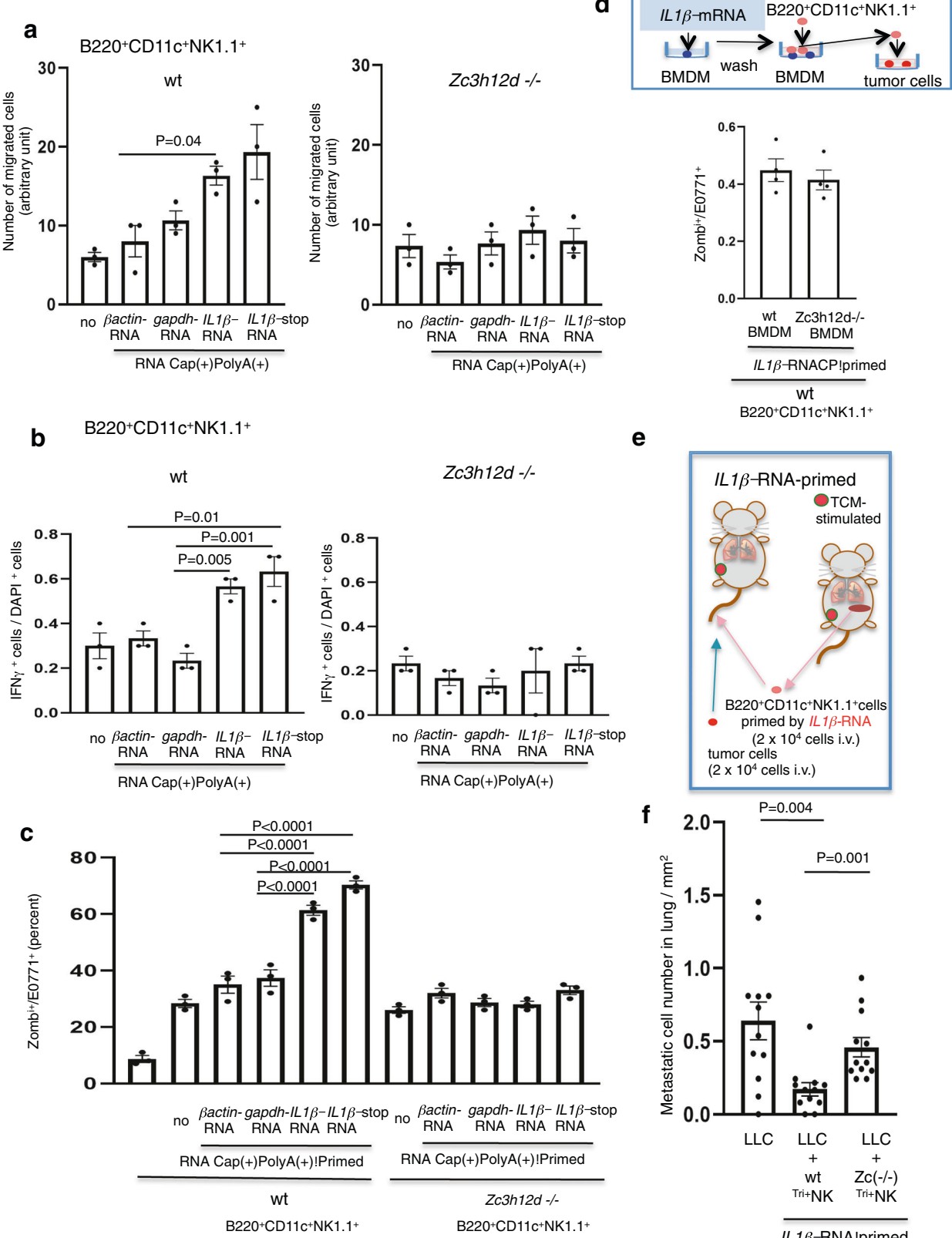

recognition ability was adopted in this protein to regulate RNA based on their structures[57] during branching between ZC3H12D and other Regnase family genes.

The premetastatic and metastatic microenvironments contain abundant proinflammatory proteins and exRNA[7]. In tumor growth and metastasis, tumor-associated RNA packaged in EVs are nonspecifically incorporated into the tumor and host cells, usually resulting in greater malignancy in cases with suppressive phenotypes[58,59]. The mechanism of this cytoplasmic recognition of RNA has been reported to involve EVs entering the cytoplasm of cells and the RNA they contain being recognized by sensor molecules, such as TLR3 and RIG-1[60,61]. This study revealed that

**Fig. 6 ZC3H12D+NK cells harboring *IL1β*-mRNA in their nucleus function in antimetastasis. a** Migration of B220+CD11c+NK1.1+ cells derived from wild-type and *Zc3h12d–/–* mice to 10 ng/mL *βactin*-mRNA, *gapdh*-mRNA, *IL1β*-mRNA, and *IL1β-stop*-mRNA ($n = 3$ wells per group). **b** IFN-γ induction in B220+CD11c+NK1.1+ cells from TCM-stimulated wild-type and *Zc3h12d–/–* mice after incubation with *IL1β*-mRNA and *IL1β-stop*-mRNA. The IHC quantitative ratio of IFN-γ-positive cells ($n = 3$ wells per group). **c** Tumoricidal assay in vitro after coincubation of tumor cells with 10 ng/mL mRNA-primed B220+CD11c+NK1.1+ cells. The ratio of rhodamine-labeled E0771 cells that became Zombie+ dead cells in the assay. The dead E0771 cells increased after coculture with *IL1β*-mRNA- and *IL1β-stop*-mRNA-primed B220+CD11c+NK1.1+ cells from TCM-stimulated wild-type mice rather than cells from *Zc3h12d–/–* mice ($n = 3$ wells per group). **d** Tumoricidal assay system. Tumoricidal assay in vitro after coincubation of tumor cells with B220+CD11c+NK1.1+ cells after coculture with BMDMs primed with *IL1β*-mRNA (top). To exclude the direct effects of *IL1β*-mRNA on NK cells, BMDMs were washed with basal medium after priming. The ratio of rhodamine-labeled E0771 cells that became Zombie+ dead cells in the assay (bottom, $n = 4$ wells per group). **e** Antimetastatic assay system in vivo. *IL1β*-mRNA-primed B220+CD11c+NK1.1+ cells of TCM-stimulated mouse spleens and then applied to another TCM-stimulated mouse. Rhodamine-labeled tumor cells were injected to assess metastasis. **f** Metastatic LLC cell numbers in lungs with or without pretreated *IL1β*-mRNA-primed B220+CD11c+NK1.1+ cells from wild-type and *Zc3h12d–/–* mice are shown ($n = 12$ mice per group). B220+CD11c+NK1.1+ is shown as ^TriNK. In the graphs, averages ± SEM and the results of Student's *t*-test (two-sided) or one-way ANOVA with Bonferroni correction are shown. The *P* values are shown in the figure. Source data are provided as a Source Data file.

**a** ZC3H12D column captured *IL1β*-mRNA by simply passing it through tissue CM and TCM without any process for dissolving EVs. Also, ZC3H12D protein beads were able to capture *IL1β*-mRNA in mouse serum. These indicated that "uncaged" free mRNA exists in CM and that they have direct interactions with ZC3H12D protein on the cell surface.

This study demonstrates for the first time that mRNA is itself utilized as an intercellular messenger molecule with nuclear functions. When mRNA is in a subnuclear location, it is expected to have a different function from translation dictated by the central dogma. Mutant mRNA with multiple stop codon mutations also elicited the same phenomenon, indicating that the exotic mRNA functions in a translation-independent manner. In the model in Fig. 8, ZC3H12D in NK cells may regulate nuclear function with tumoricidal effects. Interestingly, some gene transcriptomic signatures that are common between human and mouse NK cells[62], including *IL1rn* and *Dusp1*, were upregulated in the nuclear fraction. Dusp1 is a key negative regulator of mitogen-activated protein kinases and directly linked to cell survival by attenuating cellular stresses[39,40,63]. In the experiments, *IL1β*-mRNA entrained in the nucleus seems to be bound to RNA Pol II. These data indicate that mRNA transcription of some genes may be regulated by transported nex-mRNA in the nucleus. Interestingly, some of the ZC3H12D+NK cells possessing *IL1β*-mRNA showed the irregular shape of the nucleus, although cell death was not increased as a whole. In addition, IFN-γ production levels in NK cells were not *IL1β*-mRNA dose dependent. These data imply that a relatively small amount of *IL1β*-mRNA is enough to elicit antitumor activity in NK cells. Therefore, it is speculated that the nex-mRNA-ZC3H12D-axis in B220+CD11c+NK1.1+NK cells suppressed lung metastasis in an adoptive immunotherapy mouse model.

CD56^bright NK cells are assumed to be the human counterpart of murine B220+CD11c+NK1.1+ cells. Both are in lymphoid organs and effectively produce IFN-γ[31]. The ZC3H12D expression levels in CD56^dim and CD56^bright NK cells were alike. However, *IL1β*-mRNA-mediated IFN-γ production was stronger in CD56^dim than CD56^bright NK cells. In contrast, *IL1β*-mRNA induced NKG2D in CD56^bright NK cells, implying that NKG2D-dependent antitumor responsiveness was enhanced in cells[64]. Together, *IL1β*-mRNA may induce tumoricidal activity on CD56^dim and CD56^bright NK cells in vivo.

## Methods
**Reagents**. The following primary antibodies and factors were used in this study: anti-ZC3H12D antibodies for mouse (Abcam, ab1000862, dilution 1:100–200), E14 (Santa Cruz Biotechnology, sc-169839, dilution 1:100–200) and M15 (Santa Cruz Biotechnology, sc-169840, dilution 1:100) for fluorescence-activated cell sorting (FACS) analysis, and anti-ZC3H12D (Proteintech, 24991-1-AP, dilution 1:100) for western blotting. Anti-ZC3H12D antibodies for human (Abcam, ab1000862,

dilution 1:100), anti-ZC3H12A antibody for human (N3C3, GeneTex, GTX110807, dilution 1:100), and histone H2AXS139ph (phospho-Ser139) antibody (GT2311, GeneTex, GTX628789, dilution 1:200) were used in IHC analyses. For mouse cell sorting and FACS analyses, the following were used. For mouse CD11c: PE anti-mouse CD11c (BioLegend, 117308 dilution 1:100–200); isotype control, PE Armenian hamster IgG (eBioscience, 12-4888-81, dilution 1:100); Alexa Fluor 647 anti-mouse CD11c (BioLegend, 11732, dilution 1:100); and isotype control, Alexa Fluor 647 Armenian hamster IgG (BioLegend, 400526, dilution 1:100). For mouse NK1.1: Brilliant Violet 421 anti-mouse NK-1.1 (BioLegend, 108741, dilution 1:100); isotype control, Brilliant Violet 421 mouse IgG2a, κ (BioLegend, 400260, dilution 1:100); APC anti-mouse NK-1.1 antibody (BioLegend, 108710, dilution 1:100); and isotype control, APC mouse IgG2a, κ (BD Pharmingen, 550882, dilution 1:100). For B220: PE/Cy7 anti-mouse/human CD45R/B220 antibody (BioLegend, 103222, dilution 1:200); isotype control, PE/Cy7 rat IgG2a, κ (BioLegend, 400522, dilution 1:200); APC anti-mouse/human CD45R/B220 antibody (BioLegend, 103212, dilution 1:200); and isotype control, APC Rat IgG2a, κ (BD Pharmingen, 554690, dilution 1:200). For CD45: PE/Cy7 anti-mouse CD45 antibody (BioLegend, 304014, dilution 1:200) and isotype control, PE/Cy7 rat IgG2b (BioLegend, 400127, dilution 1:200). In cell sorting for human CD56: PE anti-human CD56 (NCAM) antibody (BioLegend, 362524, dilution 1:50–100) and isotype control, mouse IgG1, κ (BioLegend, 400111, dilution 1:50–100). In cell sorting for human CD3: Alexa Fluor 488 anti-human CD3 antibody (BioLegend, 300454, dilution 1:100) and isotype control, mouse IgG1, κ (BioLegend, 400132, dilution 1:100). Anti-IFN-γ antibodies for mice (D-17, Santa Cruz Biotechnology, sc-9344, dilution 1:200), and humans (B27, BioLegend, 506501, dilution 1:200) were used in cell staining. hIL2 (BioLegend, 791902) and hIL12 (Proteintech, HZ-1256) were used in the cell culture. Anti-SC35 (Abcam, ab204916, dilution 1:100), anti-SFPQ (C23, MBL, RN014MW, dilution 1:1000), anti-PML (MBL, K0196-3, dilution 1:1000), anti-fibrillarin (38F3, Biolegend, 905001, dilution 1:500), and Alexa Fluor 647 anti-NKG2D (Biolegend, 320825, dilution 1:100) were also used in cell staining. Anti-DNA polymerase II (F-12, Santa Cruz Biotechnology, sc-55492, dilution 1:100) was used for immunoprecipitation. In FACS analysis, anti-mouse CD16/CD32 monoclonal antibody (BD Pharmingen, 553142, dilution 1:50) or human Fc receptor blocking solution (BioLegend, 422301, dilution 1:20) were used to block nonspecific antibody bindings.

**Animals**. C57BL6 mice were purchased from Clea Japan (Tokyo, Japan) or SLC (Shizuoka, Japan). *Zc3h12d–/–*[20] and *Zc3h12a–/–*[34] were used in the experiments. All animal procedures were performed according to the guidelines of the Animal Care and Use Committee of Shinshu University and Tokyo Women's Medical University, after the approval of a protocol by the Committees. Mice were maintained under a constant temperature (22 °C) and humidity (50%), and a 12-h light/dark cycle.

**Tumor cell lines and TCM**. For mouse tumor cell lines, E0771 breast cancer cells (American Type Culture Collection (ATCC), CRL-3461) were originally established by Dr Sirotnak (Memorial Sloan-Kettering Cancer Center, New York, NY, USA) and provided by Dr Mihich (Roswell Park Memorial Institute, Buffalo, NY, USA)[65]. LLC (RCB0558), RAW264.7 (RCB0535), THP1 (RCB1189), 293T (RCB2202), and B16 melanoma cells (RCB1283) were purchased from Riken BRC. LLC (3LL) cells (JCRB Cell Bank, JCRB1348) and the 3LL in vivo passage line were supplied by the Japanese Foundation for Cancer Research (Tokyo, Japan). Mouse cells were expanded and prepared as stocked samples and used during the passages for less than 2–3 months. Cells were maintained in Dulbecco's modified Eagle's medium (DMEM) supplemented with 10% fetal bovine serum (FBS), 100 units/mL penicillin G sodium, and 100 μg/mL streptomycin sulfate. Human breast cancer cells MCF7 (HTB-22), MDAMB231 (HTB-26), T47D (HTB-133), and SKBR3 (HTB-30) were obtained from the ATCC and passaged in Shinshu University. HCC1954 (CRL-2338) cells and 786-O (CRL-1932) cells were purchased from the ATCC. Human cells were cultured in DMEM/Ham's F-12 medium. The cell lines

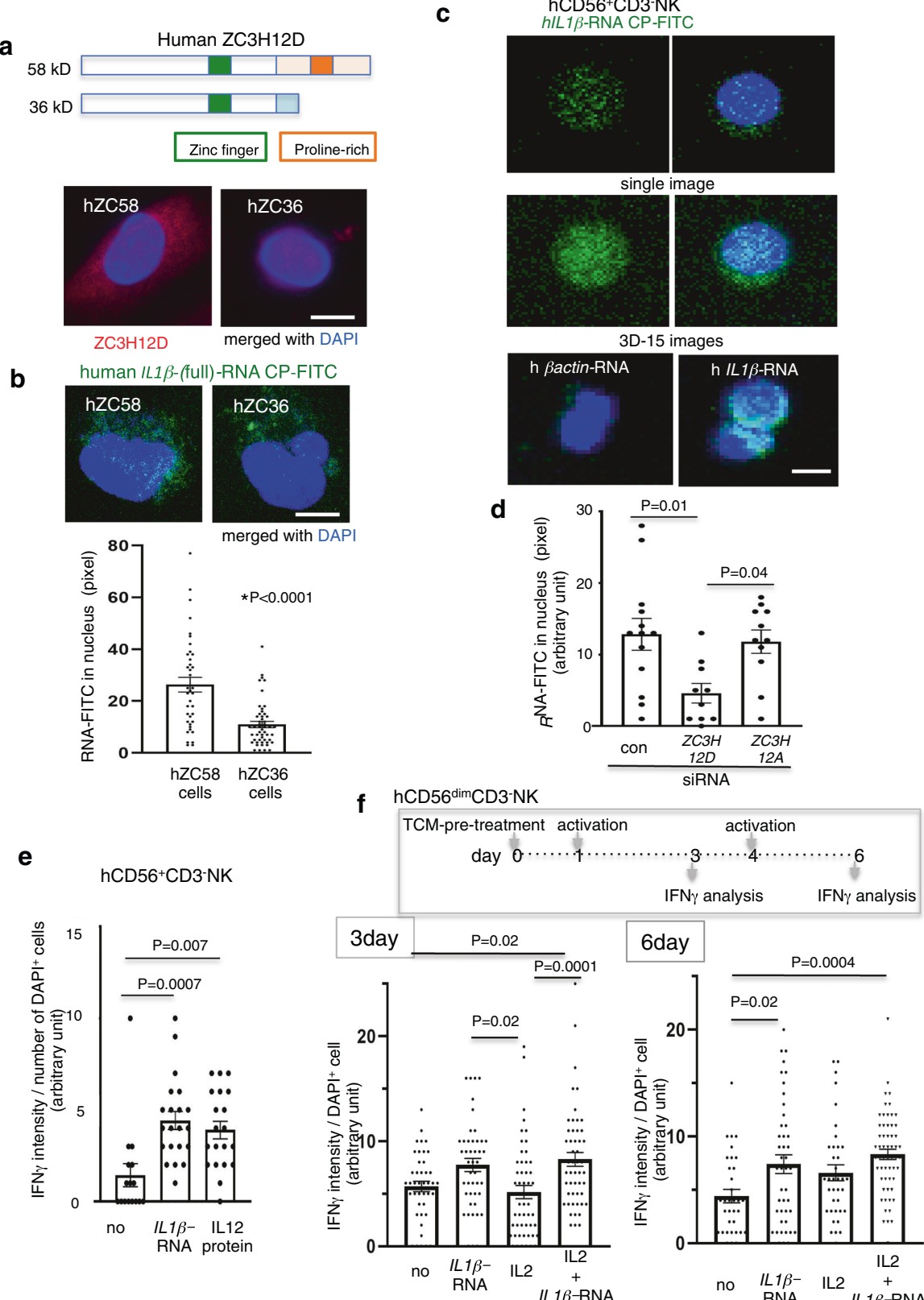

were tested negative for mycoplasma contamination by using MycoAlert Myco-plasma Detection Kit (Lonza, LT-07-218). Cells were not authenticated. TCM was obtained by incubating cells overnight in a serum-free medium without cell death. Gathered TCMs were centrifuged at 215 × g for 5 min, and the supernatant was further centrifuged at 21,500 × g for 10 min. TCMs were then stored at −80 °C before experiments.

**TCM and CD45⁺ cell culture system using organ tissues**. In the organ culture experiments, 2 mm² lung or liver tissue specimens were serum-free cultured overnight in high glucose DMEM with gentamicin. After collection of the culture media, samples were first centrifuged at 215 × g for 10 min, and the supernatant was further centrifuged at 21,500 × g for 30 min at 4 °C. For the coculture system of tissue and CD45 cells, lungs were incubated in DMEM with FBS (1%) in an upper

**Fig. 7 hZC3H12D contributes to the uptake of nex-h*IL1β*-mRNA with function. a** Scheme for two splicing isoforms of hZC3H12D. The long and short forms encode 58 and 36 kDa proteins, respectively. Both have a common N-terminal sequence, including a ZF domain (green), and the long form has a proline-rich domain (orange; top). IHC analysis of ZC3H12D in the long form (hZC58)- and short form (hZC36)-overexpressing 786-O cells (bottom). Experiments were repeated twice with similar results. **b** Representative confocal microscopy images of h*IL1β*-mRNA-FITC uptake in the hZC58 or hZC36 cell nucleus (top). Quantitative signals of h*IL1β*-mRNA-FITC in the hZC58 or hZC36 cell nucleus (bottom; 3D images per group: $n = 38$ for hZC58 and $n = 50$ for hZC36). **c** Uptake of h*IL1β*-mRNA-FITC into CD56$^+$CD3$^-$NK cell nucleus 3 h after applying 50 ng/mL RNA with 200 U/mL IL-2 protein. Confocal microscopy imaging showing the DAPI-stained nucleus in single images (top) and 3D stacks combining 15 images from top to bottom (middle). 3D stack images show a comparison of nuclear uptake between h*IL1β*-mRNA and h*βactin*-mRNA (bottom). **d** Nuclear signals of h*IL1β*-mRNA-FITC in CD56$^+$CD3$^-$NK cells treated with siRNA. Control (con) is a nontargeting siRNA ($n = 13$, 10, and 11 cells for control, *ZC3H12D*-siRNA and *ZC3H12A*-siRNA, respectively. Each cell image is composed of 15-stacked 3D images). **e** IHC analysis of IFN-γ induction 48 h after RNA application with 200 U/mL IL2 to CD56$^+$CD3$^-$NK cells. The positive control was incubated with 20 ng/mL IL12 protein in an IL2 culture medium (IFN-γ signals from $n = 16$, 21, and 20 ZC3H12D+ cells for no, *IL1β*-RNA, and IL2 protein, respectively). **f** IHC analysis of IFN-γ induction 3 and 6 days after stimulation of h*IL1β*-mRNA and IL2 protein for MDAMB231-TCM-pretreated CD56$^{dim}$CD3$^-$NK cells. The assay system is shown (top). Two donor NK cells were independently used. no, control basal culture medium (bottom left, $n = 42$, 45, 51, and 51 cells for no, *IL1β*-RNA, and IL2 protein, and *IL1β*-RNA plus IL2 protein, respectively). Bottom right, $n = 34$, 44, 37, and 60 cells for no, *IL1β*-RNA, and IL2 protein, and *IL1β*-RNA plus IL2 protein, respectively). Bars, 5 μm. In the graphs, the averages ± SEM, and the results of Student's *t*-tests (two-sided) or one-way ANOVA with Bonferroni correction are shown. The *P* values are shown in the figure. Source data are provided as a Source Data file.

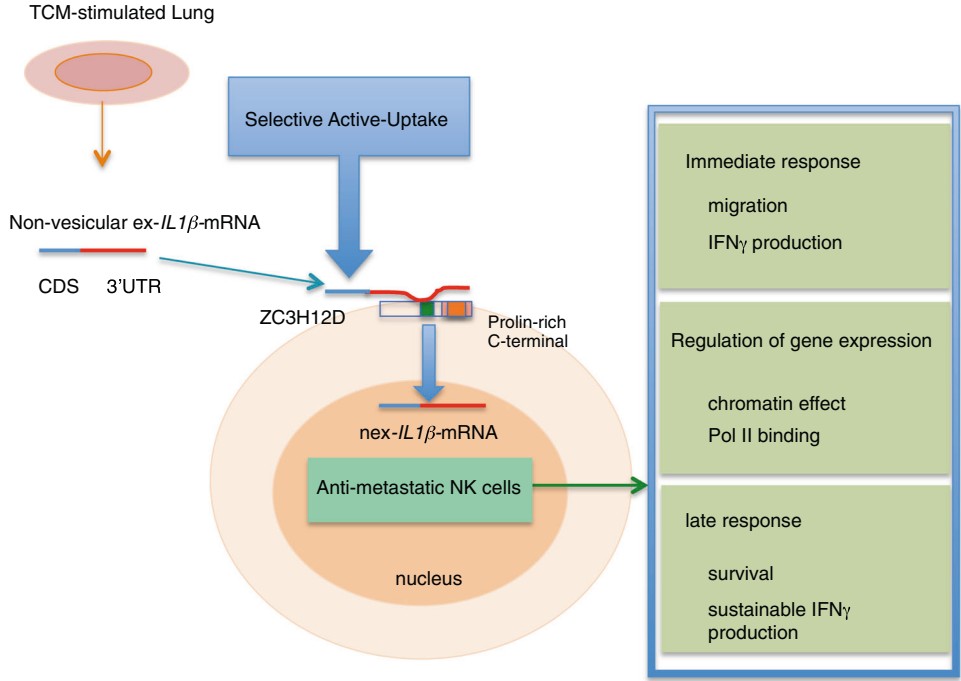

**Fig. 8 Model of the translation-independent function of nex-mRNA.** The tumor-associated nex-*IL1β*-mRNA 3'-UTR was trapped by ZC3H12D protein on the NK cell surface and introduced to the nucleus via its C-terminal containing a proline-rich domain. Then, this may regulate antiapoptotic gene expression associated with RNA Pol II and control NK cell migration, survival, and sustainability of IFN-γ production.

well culture insert (400 nm pores). CD45$^+$ cells isolated from the lungs were cultured in the lower wells, which were stimulated by organ tissues in the upper wells for 24–48 h.

**Primary culture of human cells.** Primary human lung microvascular cells (Applied Cell Biology Research Institute, ACBRI 468) were cultured in EBM™-2 Microvascular Endothelial Cell Growth Medium-2 SingleQuots™ Supplements and Growth Factors (Lonza). After becoming subconfluent in a 10 cm dish, human TCMs were added at 15–20% and incubated for 24 h. The culture medium was then gathered as lung EC-CMs. Primary hPBMCs (Lonza CC-2702) were cultured in LGM-3 medium (Lonza) with 200 IU/mL IL2. Cells were received from full consent donors with IRB-approved protocols.

**Experimental metastatic model.** B220$^+$CD11c$^+$NK1.1$^+$ cells derived from TCM-stimulated mouse spleens were primed by *IL1β*-RNA for 16–18 h, and $2 \times 10^4$ of these cells per mouse were intravenously (i.v.) injected into other TCM-stimulated mice. Then, to examine the metastatic tumor cell homing, a $2 \times 10^4$ fluorescent dye (PKH26; Sigma-Aldrich, St. Louis, MO, USA)-labeled metastatic cells were i.v. administered to pretreated mice. Forty-eight hours after tumor cell infusion, the lungs were perfused with phosphate-buffered saline (PBS) under physiological

pressure to exclude circulating tumor cells. Six lung tissue fragments (3 mm in diameter) were randomly excised, and four 10-μm sections per fragment were examined under a confocal (LSM-710; Carl Zeiss MicroImaging GmbH, Germany) or fluorescent microscope (BZ-9000; Keyence, Osaka, Japan). The labeled tumor cell counts were normalized to the total tissue surface area. Age- and sex-matched littermates were used for the experiments.

For the spontaneous metastatic assay, E0771 cells were macroscopically detected in the lungs of tumor-bearing mice after primary tumor resection. Syngeneic tumor grafts were generated via subcutaneous or mammary fat pad implantation of $5 \times 10^6$ tumor cells into 7- to 8-week-old mice.

**Flow cytometry, cell sorting, and cell staining.** To collect B220$^+$CD11c$^+$NK1.1$^+$ cells from TCM-stimulated wild-type, *Zc3h12d*+/−, and *Zc3h12d*−/− mouse spleens, cells were purified with a cell sorter (MoFlo Astrios$^{EQ}$ (Beckman coulter) and FACSAria™ III (BD Bioscience)). B220$^+$CD11c$^+$NK1.1$^+$ cells derived from wild-type and *Zc3h12a*$^{−/−}$ mice were not prestimulated by TCM. Human CD56$^+$CD3$^-$NK cells were obtained by FACSAria™ III. Mouse livers and lungs were digested with 0.5 mg/mL collagenase, 1 mg/mL dispase, and DNase at 37 °C for 45 min to obtain B220$^+$CD11c$^+$NK1.1$^+$ cells. Cell suspensions were incubated with mouse CD45-microbeads (MACS; Miltenyi Biotec, Auburn, CA, USA), and the

captured cells were used for in vitro culture in Supplementary Fig. S1. For the detection of ZC3H12D in mouse PBMCs in Fig. 1b, staining was carried out as described below, and cells were analyzed using a flow cytometer (Cytomics FC500, Beckman Coulter). For the staining of the cell surface, PBMCs were gathered using Ficoll-Paque (Histopaque®-1083; Sigma-Aldrich) and/or hemolysis and stained with 0.5 μg antibodies/$10^6$ cells in 100 μL volume. For intracellular staining, cell fixation and permeabilization (Fixation/Permeabilization Kit; BD Pharmingen) treatments were administered before the addition of antibodies. BD FACSCelesta (BD Bioscience) was also used for flow cytometry, and data analyses were done by using FACSDiva v8 (BD Bioscience) and FlowJo v7.6 (FlowJo LLC).

**Immunoprecipitation**. To crosslink protein-RNA complexes, cells cultured in a 10 cm dish were washed with PBS twice and received 254 nm UV irradiation (1500 J/$m^2$) before protein extraction. Cells were dispersed in 10 mM HEPES (pH 7.6), 15 mM KCl, 2 mM MgCl$_2$, and 0.1 mM EDTA with protease inhibitors (G6521; Promega). NP-40 was then added to give a final concentration of 0.2%. After initial centrifugation (600 × g for 1 min), the supernatant was further centrifuged (21,500 × g for 15 min) to obtain nuclear fraction. Nuclear proteins were extracted with RIPA buffer [50 mM HEPES (pH 7.6), 150 mM NaCl, 0.5% Triton X-100, and 0.5% sodium cholate] with protease inhibitors (G6521; Promega) and RNase OUT (Thermo). An antibody preincubated with protein G magnetic beads (Thermo) was added to the extract in the presence of RQ1 RNase-Free DNase (Promega) and incubated for 1 h at 4 °C. The magnetic beads were washed with Tris-buffered saline + Tween-2 (TBST) four times and further washed with TBST + 1 M NaCl. Then, the beads were incubated with protease K (1 h at 37 °C). Then, phenol-chloroform extraction and ethanol precipitation were performed.

**Uptake of RNA**. Before RNA application, primary cells were incubated in 1% FBS culture medium for 1 h, and the cell lines were preincubated with or without 1% FBS for 3 h. Human NK cells were not starved. To quench the RNA uptake, 4% paraformaldehyde was added in 5 min (1% final concentration) and washed three times. For the intracellular staining of ZC3H12D, 0.1% Triton X-100 was used before staining. To obtain the confocal z-stack images (Leica TCS SP8 confocal microscopy), the top and bottom of the nucleus were orientated by the 4′,6-diamidino-2-phenylindole (DAPI) signal, and a series of 15 images from the top to the bottom of the nucleus were captured and constructed into 3D images for a cell. For accurate comparisons among samples, image data were acquired within 2 days. Image data were analyzed by Leica Application Suite X (LAS X v3, Leica). All cells were checked with a single image of the central portion of the nucleus to confirm the nuclear RNA uptake.

**Migration assay**. The migration of cells was evaluated using a chemotaxis Boyden chamber (Neuro Probe). The top and bottom wells were separated by a 5-μm pore-size polyvinylpyrrolidone-free polycarbonate filter (Nucleopore; Costar). Various RNAs were applied to the bottom wells. An aliquot (50 μL) of the cell suspension (2 × $10^5$–2 × $10^6$ cells/mL) was seeded in each of the top wells and incubated for 3.5 h.

**IFN-γ induction and tumoricidal assay**. B220$^+$CD11c$^+$NK1.1$^+$ cells were stained using an anti-mouse IFN-γ antibody 16 to 24 h after incubation with 10 ng/mL m*IL1β*-RNA. Human CD56$^+$CD3$^-$NK cells were stained with an anti-human IFN-γ antibody 48 h after priming with 50 ng/mL h*IL1β*-RNA in an LGM-3 medium (Lonza) supplemented with 200 IU/mL IL2. For the positive control of IFN-γ induction, cells were incubated with 40 ng/mL hIL-12 for 48 h. The Zombie Green Fixable Viability Kit (BioLegend) was used to detect the dead PKH26-stained tumor cells 24 h after incubation with B220$^+$CD11c$^+$NK1.1$^+$ cells that were primed with 10 ng/mL m*IL1β*-RNA for 16–18 h.

**Real-time apoptosis and necrosis assay**. ZC$^+$RAW, B220$^+$CD11c$^+$NK1.1$^+$NK, and hZC58 cells were traced after the application of *IL1β*-mRNA or *βactin*-mRNA. To detect cell death, RealTime-Glo™ Annexin V Apoptosis and Necrosis Assay (Promega) are used. In this assay, Annexin V binding is detected with a luminescence signal, and necrosis is detected with a fluorescence signal.

**Vector construct and establishment of ZC3H12D stably expressing cells**. hZC3H12D expression vectors (hZC58 and hZC36), in which the hZC3H12D CDS is cloned into a pCMV6-entry vector (C-terminal myc-FLAG tag), were purchased from OriGene Technologies, Inc. (Rockville, MD, USA). mZC3H12D was PCR-amplified using the primer set 5′-GGTACCATGGAGCATCGGAGCAAGATGG-3′ and 5′-CTCGAGTTAAGGATCCCCCAACGGAGCACC-3′ and then cloned into a pCR Blunt II vector (Thermo). The cloned fragment was double-digested by *Kpn*I-*Xho*I and subcloned into pcDNA3 attached with a C-terminal FLAG tag. To establish mZC3H12D or hZC3H12D stably expressing cell lines, one of the aforementioned constructs was transfected into a cell line, and cells were cultured for more than 2 weeks in the presence of 400 μg/mL G418. After G418 selection, cell lysates were tested by western blotting with DDDDK antibody (MBL Co., Ltd, Japan, PM020, dilution 1:1000) probing to confirm the expression of ZC3H12D-FLAG proteins.

**Vector constructs for RNA synthesis**. mIL1βs were PCR-amplified (primer set for full length: 5′-AACAAACCCTGCAGTGGTTCGAG-3′ and 5′-GTTTGTTTTG TTTTAATGAAATTTATTTCATACTCATCAAAGCAATG-3′; for CDS: 5′-AAC AAACCCTGCAGTGGTTCGAG-3′ and 5′-TTAGGAAGACACGGATTCCATGG TGAAG-3′; for UTR: 5′-AGTATGGGCTGGACTGTTTCTAATGC-3′ and 5′-GT TTGTTTGTTTTAATGAAATTTATTTCATACTCATCAAAGCAATG-3′) and cloned into a pCR Blunt II vector. To prepare mIL1β nonsense mutant (*IL1β*-stop-mRNA), three codons encoding Cys (TGC for Cys34, TGC for Cys104, and TGT for Cys188) were replaced with a stop codon (TGA). hIL1β was PCR-amplified (primer set: 5′-ACCAAACCTCTTCGAGGCACAAGG-3′ and 5′-CTTCAGT-GAAGTTTATTTCAGAACCATTGAACGATGATATTTGCTC-3′) and cloned into a pCR Blunt II vector. To prepare the human-mouse chimeric IL1β construct, the *Not*I-*Nde*I fragment in the mIL1β/pCR Blunt II was replaced with a double-digested fragment excised from the hIL1β plasmid. As a result, this chimera encoded hIL1β (5′-UTR + first half of CDS; 502 bp) combined with mIL1β (second half of CDS + 3′-UTR; 843 bp) at the NdeI site. mIL1β 3′-UTR was further divided into four fragments. The following primer sets were used to amplify the fragments (first fragment: 5′-AGTATGGGCTGGACTGTTTCTAATGC-3′ and 5′-AACAGA ATGTGCCATGGTTTCTTGTG-3′; second fragment: 5′-CGGCCAAGACAGGTC GCTC-3′ and 5′-CTTGAATCAACTTAAATAGATCAACCAATCAATAAATA C-3′; third fragment: 5′-TATTTATTTTATGTATTTATTGATTGGTTGATCTATT TAAGTTGATTCAAG-3′ and 5′-TTCTCAGCTTCAATGAAAGACCTCAGT G-3′; fourth fragment: 5′-GGGATGAATTGGTCATAGCCCG-3′ and 5′-GTTTGT TTGTTTTAATGAAATTTATTTCATACTCATCAAAGCAATG-3′). The amplicons were then cloned into a pCR Blunt II vector. There fragments were 148–153 bp, and each one had at least one 50 bp overlap with its neighboring fragments. These constructs were linearized by *Spe*I digestion and used for the in vitro translation system (RiboMAX Large-Scale RNA Production System-T7; Promega, WI, USA). To prepare fluorescent-labeled RNA, Fluorescein RNA Labeling Mix (Sigma-Aldrich) was mixed in the RNA synthesis reaction. To introduce Cap modification, synthesized RNA was mixed with Vaccinia virus Capping enzyme [M2080; New England Biolabs (NEB), MA, USA] and Cap 2′-O-Methyltransferase (M0366; NEB); after the reaction, RNA was purified by phenol-chloroform extraction and ethanol precipitation. Subsequently, part of the resulting RNA was mixed with Poly(A) polymerase (M0276; NEB) to make a Poly(A) adduct. Before use, all RNAs were purified using a spin column-based purification system (PureLink RNA Mini Kit; Thermo).

**EMSA**. ssRNA oligonucleotides were purchased from Eurofins Genomics. A Biotin Labeling Kit (89818; Thermo) was used to introduce biotin labeling at the 3′-end of the oligonucleotides. To study the interaction between RNA oligonucleotides and ZC3H12D protein, a LightShift Chemiluminescent RNA EMSA Kit (20158; Thermo) was used. Purified ZC3H12D protein was mixed with biotin-labeled oligonucleotides in the presence of glycerol, reaction buffer components [final concentration: 100 mM HEPES (pH 7.3), 200 mM KCl, 10 mM MgCl$_2$, and 10 mM dithiothreitol], and tRNA supplied with the kit; the mixture was then run in a 5% to 20% gradient gel (Fujifilm Wako Pure Chemical, Japan). Next, RNAs were electrotransferred onto a nylon membrane, fixed on the membrane with UV irradiation (1200 J/$m^2$), and probed with streptavidin-horseradish peroxidase conjugate. To visualize the biotin-labeled RNA, chemiluminescent reagents were applied on the membrane, and luminescent signals were detected by a CCD imager.

**Immunoprecipitation for western blot or qPCR**. To crosslink protein-RNA complexes, cells cultured in a 10 cm dish were washed with PBS twice and received 254 nm UV irradiation (1500 J/$m^2$) before protein extraction. Cellular proteins were extracted with RIPA buffer [50 mM HEPES (pH 7.6), 150 mM NaCl, 0.5% Triton X-100, and 0.5% sodium cholate] with protease inhibitors (G6521; Promega). Anti-DYKDDDDK tag antibody magnetic beads (Fujifilm Wako Pure Chemical) were added to the lysate and incubated for 1 h at 4 °C. The magnetic beads were washed with TBST four times and further washed with TBST + 1 M NaCl. For western blotting analysis, washed beads were mixed with sodium dodecyl sulfate-containing sample buffer and heated at 95 °C. To retrieve RNA, beads were treated with RQ1 RNase-Free DNase (Promega), and after a TBST wash, they were incubated with protease K (1 h at 37 °C). Then, phenol-chloroform extraction and ethanol precipitation were performed.

**RNA preparation from cytoplasmic and nuclear fractions**. Cells were dispersed in 10 mM HEPES (pH 7.6), 15 mM KCl, 2 mM MgCl$_2$, and 0.1 mM EDTA with protease inhibitors (G6521; Promega), and NP-40 was added to give a final concentration of 0.2%. After initial centrifugation (600 × g for 1 min), the supernatant was further centrifuged (21,500 × g for 15 min), and an RNA-containing pellet was obtained by phenol-chloroform extraction and ethanol precipitation. After 600 × g centrifugation, the precipitating nuclear fraction was mixed with Trizol (Thermo) to isolate RNA.

**Extraexosome and exosome fractions from TCM and lung TCM**. TCMs and lung CM that serum-free cocultured with or without TCMs (lung TCM/CM) were centrifuged 300 × g for 5 min, and the supernatants were further centrifuged 3000 × g for 10 min and filtered. Exosomes can be fractionated by size-exclusion

chromatography, ultracentrifugation, magnetic bead pull-down assay, and polymer precipitation[66]. Among them, ExoQuick-TC (System Biosciences), a polymer used to precipitate exosomes, is the best method to eliminate exosomes from CM.

The samples were treated by ExoQuick-TC (System Biosciences) overnight and then centrifuged 1500 × g for 30 min. The supernatants were further centrifuged 20,000 × g. The resulting supernatants were treated by RNAiso Blood (Takara Bio) to obtain RNA as extraexosome RNA. The exosome pellets were washed by PBS and treated by Trizol reagent (Thermo Fisher Scientific) to isolate exosome RNA. All procedures were carried out at 4 °C.

**RNA pull-down assay by ZC36 protein beads from mouse serum.** Mouse blood was taken from the heart, and serum was separated using Microtainer tubes (Gold; Becton Dickinson). Then, serum was mixed with ZC36-protein-anti-DYKDDDDK magnetic beads for 1 h at 4 °C. Beads were washed with PBS three times, and RNA attaching to the beads was isolated by Trizol reagent.

**Protein purification and protein column.** The hZC36 expression vector (hZC36/pCMV6-Entry) was transfected into HEK293T cells. Forty-eight hours after transfection, cells were harvested and stored at −80 °C. RIPA buffer with protease inhibitors was added to the frozen cells, and the lysate was centrifuged at 21,500 × g for 15 min. The supernatant was 0.22 μm filtered and loaded on a column holding anti-DYKDDDDK tag beads (Fujifilm Wako Pure Chemical). Then, the column was washed with PBS (10 bed volume), PBS + 1 M NaCl (10 bed volume), and PBS (10 bed volume). This column was used as a ZC3H12D protein column to absorb substances capable of binding ZC3H12D protein. ZC3H12D protein on the column was eluted by PBS, 0.5 M NaCl, and 100 ng/mL DYKDDDDK peptide (Fujifilm Wako Pure Chemical), and the effluent was concentrated with an Amicon Ultra centrifugal filter unit (Merck, NJ, USA). The affinity column-purified ZC3H12D protein was further purified using a Q-Sepharose HP column attached to a chromatography system (GE Healthcare, NJ, USA). ZC protein diluted in 50 mM HEPES (pH 7.4) was loaded on a Q-Sepharose HP column and eluted with a NaCl gradient (0–1 M NaCl). Q-Sepharose-purified ZC3H12D protein was used for FCS. First, equal molar Alexa Fluor 488 NHS ester (Thermo) was mixed with ZC3H12D protein and incubated for 2 h at room temperature. After the labeling reaction, ZC3H12D protein was purified using the Q-Sepharose HP column again. The purified sample was concentrated for storage and diluted in 50 mM HEPES (pH 7.4) and 100 mM NaCl buffer before FCS measurements.

**FCS.** FCS measurements were performed using a home-built confocal microscope setup. The excitation light was obtained from a diode laser (OPG-1000PL-488; Tama Electric, Hamamatsu, Japan) with a wavelength of 480 nm and output power of 100 μW. The laser light was coupled to a single-mode optical fiber to shape the spatial intensity profile. The output from the fiber was introduced to a collimator, expanded by two lenses, and introduced to a water immersion objective (Plan Apo VC, 60×, NA 1.20; Nikon, Tokyo, Japan) by a dichroic mirror (MD499; Thorlabs, NJ, USA). For measurements, a 100 μL sample solution was placed on a glass-based dish (3971-035; Iwaki, Shizuoka, Japan), and the focal point of the objective was adjusted to 50 μm inside the sample from the inner surface of the glass. The fluorescence photons from the sample molecules were collected by the same objective used to focus the excitation light and imaged by an achromatic lens, with a 50-μm pinhole used to eliminate background light. The photons that passed through the pinhole were collimated and imaged on a hybrid photodetector (HPD; H13223-40; Hamamatsu Photonics, Hamamatsu, Japan) by two achromatic lenses. The output signal from the HPD was preamplified by high-speed inverting amplifiers (C10778; Hamamatsu Photonics) and then recorded and analyzed by a time-correlated single-photon counting module (SPC-130-EM; Becker & Hickl, Berlin, Germany). The autocorrelation functions were accumulated for an acquisition time of 600 s and recorded using Becker & Hickl software (SPCM). Before collection of each data set, calibration of the confocal volume was performed by measuring the diffusion time of a 0.1 nM aqueous solution of rhodamine 110.

The observed autocorrelation functions in time, which ranged from 1 μs to 1 s, were analyzed using Igor Pro v7 (WaveMetrics, OR, USA). Initially, the autocorrelations obtained for Alexa Fluor 488-labeled ZC3H12D protein were fitted using a 3D diffusion model with one diffusing component and the amplitude caused by the triplet state. The autocorrelations obtained for the other samples were globally fitted using a 3D diffusion model with two diffusing components and the triplet state amplitude based on the assumption that the diffusion time of each component was constant across different conditions. The hydrodynamic radii of the samples were calculated by referring to the measured diffusion time for rhodamine 110, for which the diffusion coefficient in water was reported.

**IHC and cell counts.** Anti-ZC3H12D (ab1000862; Abcam) antibody was used to stain spleen cells, B220+CD11c+NK1.1+ cells, ZC+RAW cells, hZC36 and hZC58 cells, CD56+CD3-NK cells, and frozen tissue sections. In Supplementary Fig. S1, the immunostained cell area values are shown as the number of pixels normalized to the DAPI signal. Labeled tumor cells were detected by confocal or fluorescence microscopy and normalized by the total surface area.

**Knockdown using siRNA.** For knockdown of ZC3H12D and ZC3H12A, CD56+CD3-NK cells were treated with 300–1000 nM siRNA [ON-TARGET plus hZC3H12D (340152) and ZC3H12A (80149) siRNA (SMARTpool)] using electroporation with an Amaxa Nucleofector system (Lonza). The control siRNA was ON-TARGET plus Nontargeting pool (Dharmacon Horizon Discovery). The efficiency of knockdown at the protein level was examined with IHC staining. An RNA uptake assay was carried out 27 h after the application of 300 nM siRNA.

**Microarray analysis.** Microarray screening of spleen tissues from TCM-stimulated wild-type and Zc3h12d–/– mice was performed using a GeneChip® Clariom D Assay, Mouse Transcriptome Array 2.0 (Affymetrix). The microarray data of B220+CD11c+NK1.1+ cells from liver and lung tissue obtained using a GeneChip Mouse Genome 430 2.0 Array (Affymetrix) were previously reported[16]. Microarray data were deposited at the National Center for Biotechnology Information: Supplementary Table S1, GSE76235; Supplementary Table S2, GSE161219; Supplementary Tables S3 and S4, GSE104002.

**qPCR.** exRNA samples were isolated from 5 to 10 mL TCM, lung TCM, liver TCM, and human lung EC-CMs using RNAiso Blood (Takara) with Ethachinmate (Nippon Gene). Total RNA samples were isolated from cells using Trizol reagent (Invitrogen, Carlsbad, CA, USA) and used to generate cDNA with reverse transcriptase (SuperScript VILO, Invitrogen). For the reverse transcription with GSP, SuperScript III reverse transcriptase (Invitrogen) was used. cDNA from exRNA was amplified using TaqMan PreAmp Master Mix (Applied Biosystems, Foster City, CA, USA) before qPCR analysis. qPCR was performed using SYBR Green Master Mix or TaqMan Fast Advanced Master Mix (Applied Biosystems) in a detection system (StepOnePlus; Applied Biosystems). The gene expression levels were calculated from Ct values, and the relationship between the Ct value and the logarithm of the copy number of the target gene was confirmed to be linear using serial dilutions of the corresponding isolated DNA as a standard. In addition, gene expression levels of nuclear and cytoplasmic samples were normalized to that of Hotair and βactin, respectively. The primer sequences and probes are listed in Supplementary Table S5.

**Colocalization and statistical analyses.** For colocalization analysis, Image J v1.52 (National Institute of Health) with coloc2, a plugin for colocalization analysis (https://imagej.net/Coloc_2), was used. Data are expressed as mean ± standard error of the mean. Statistical evaluation was conducted as indicated. $P < 0.05$ was considered statistically significant. All Student's $t$-tests were unpaired and two-sided. Analysis of variance was always employed with Bonferroni's correction. $P$ value calculations were carried out by using Prism v8 software (GraphPad Software, San Diego, CA, USA). The experiments were repeated more than twice for representative images.

**Reporting summary.** Further information on research design is available in the Nature Research Reporting Summary linked to this article.

## Data availability

No restrictions on data availability. Microarray data [Supplementary Table S1, GSE76235; Supplementary Table S2, GSE161219; Supplementary Tables S3 and S4, GSE104002] are deposited in the Gene Expression Omnibus (GEO) database. Other data are available from the corresponding author upon request. Source data are provided with this paper.

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

## Acknowledgements
We thank Drs Kensuke Miyake (The University of Tokyo) and Yasuhiro Furuichi (Gene-Care Research Institute) for helpful discussions. This study was supported by Japan Society for the Promotion of Science (JSPS) KAKENHI Grant-in-Aid for Scientific Research (B) 19H03500 (S. H.) and (C)20K06554 (T. Tomita) as well as The Yasuda Medical Foundation (S. H.). This work was performed in part under the Collaborative Research Program of Institute for Protein Research, Osaka University, CR-18-05 (T. Tomita).

## Author contributions
T. Tomita and S. H. designed the study and analysis. Experiments were performed by T. Tomita, M. K., T. Mishima, Y. M., H. S., K. I., K. M., T. Matsui, H. O., S. T., T. Takao, N. I., T. Mino, O. T., Y. M., and S. H. Data analysis was performed by S. H.. The study was supervised by Y. M. and S. H. This manuscript was prepared by T. Tomita and S. H. with input from all authors.

## Competing interests
The authors declare no competing interests.
