## [Peer Review File · Nature Communications]

REVIEWER COMMENTS

Reviewer #1 (Remarks to the Author):

Authors report here a very interesting story that if is true will be paradigmatic for extracellular RNA function. However, with the data presented and the controls missed, I am not sure that all the central claims of this work are well supported. It is often the case that figures lack critical controls or seem incomplete (see below). I have also missed the use of a marker to delimit the cytoplasm in nucleus/cytoplasm localisation studies (e.g. Phalloidin) as the cytoplasm of the cells used appears to be small. Anyway, I think the 'killing' experiment is still missing. It would be to perform the experiments with an IL1B mRNA at physiological concentration including one or more stop codons to avoid the production of IL1B protein. This, with additional controls to show that the toll-like receptor pathway is not the cause of the observed effects (and I think this possibility is likely), would give a lot of value to the paper.

Major points

1. Page 4-5. I guess it would be good to highlight that the experiment treating cells from spleen with TCM has as a goal to show that cells that are not within the tumour environment can respond to TCM treatment. Why authors did not compared cells from wt vs ZC3H12D KO mice?
2. I don't understand the reasoning behind using spleen gene expression profile to identify the potential regulatory nex-mRNAs. Wouldn't make sense to analyse the supernatant of cultured tumours (TCM) after purification of extra vesicle RNA? Is really the analysis used adequate to identify nex-mRNA by transcriptomic profiling? Maybe I have missed something here. Moreover, I don't think I fully understand the reasoning behind the selection of IL1B mRNA as from my point of view is not fully justified here. Weren't other RNAs as upregulated as IL1B mRNA (based on Fig S4)? Was this a lucky guess or there was a reason to think that IL1B mRNA was going to be a) extracellular and extra vesicle and b) would produce a phenotype. As I said, I may have missed something here and if so, apologies for this.
3. Page 6. I am not familiar with the extra-exosomal sample preparation used here for the RNA analysis by RT-qPCR. After reading the mat and met I am still confused as I am not familiar with the kits. Maybe some sentences to clarify the purpose of each step would help. Do authors control for extra-exosomal RNA using RNase treatment (that should digest unprotected RNA)?
4. Page 6 / Figure 2a. While authors discuss an interaction between ZC3H12D and IL1B mRNA, the microscopy shows a clear exclusion of the fluorescence derived from each molecule. Since the microscopy used is expected to have low resolution (200-300 μm), if both molecules do interact, they would be expected to be present at (at least partially) overlapping areas instead of juxtaposed. Moreover, I don't understand why other FITC-labelled RNAs weren't used to show specificity and why the NoCM control is not shown in the microscopy above.
5. Fig. 2b. How is copy number estimated? Why other (control) RNAs aren't tested? Ideally this immunoprecipitation should be combined with RNAseq of the eluates to provide an overview of the RNAs bound by ZC3H12D and show that IL1B mRNA is a dominant target. Otherwise, it is difficult to justify why focusing on IL1B mRNA and not others. It is also difficult to identify positive and negative RNA controls to use in downstream experiments. Figure legends of panels d-e are not sufficiently elaborated to understand what these plots show (at least for people not familiar with them). At a glance, I can see some coherency regarding the Y axis, but a large data distribution in the X axis. Is this what authors expected?
6. Fig. 3b. A control with unlabelled IL1B mRNA is missing. The signal observed is very close to noise

levels, so this control is critical to differentiate between signal and noise. Moreover, the experiments in Figure 3 are generated with large amounts of exogenous IL1B mRNA. Authors should prove that in physiological TCM IL1B mRNA concentrations (i.e. the one found in TCM), the RNA would be detected in the nucleus.

7. Figure 3c. Not sure what is the Wt panel as authors only show Zc3h12d +/- and -/-. Without the +/+ control this experiment is inconclusive and difficult to evaluate. All the experiments in figure 3 should include an unlabelled RNA control to define the autofluorescence level. Without this, it is very difficult to assess whether the observed fluorescence is signal or autofluorescence.

8. Figure 4. Can authors show what is the affinity of Zc3h12d for the target RNA in the EMSA assay? In Figure 4e, without a splitting of the channels, the inclusion of a negative RNA control, and a paralleled Zc3h12d KO line analysis, it is not possible to fully interpret and draw solid conclusions from this microscopy experiment.

9. Figure 5. If the effect is independent of translation, I would expect to see the same effect if one or more stop codons are inserted into the IL1B mRNA to avoid the production of the protein. This would be the only way to show that the effect is translation-independent although it cannot be excluded the possibility that IL1B mRNA interaction with membrane proteins triggers a signalling cascade through, for example, RNA-binding toll-like receptors. Controls showing that toll-like receptor pathway is not activated should be shown. Unfortunately, the Cap/poly(A) – is a good control but not sufficient to rule translation and toll-like receptor roles in the observed effects. 1) An mRNA lacking cap and poly(A) can be translated if added in large quantities. 2) An mRNA lacking cap and poly(A) will exhibit lower stability as it is susceptible to exonucleases. These two factors can explain why authors see the effects when adding large amounts of cap and poly(A) less RNA. Controls using RNAs with near identical sequence but that cannot be translated are thus essential here. Also showing that other RNAs do not produce this effect when used at the very same (or even higher) concentration, would be critical to rule out RNA sensing and triggering of the antiviral programme. This criticism is applicable to Fig 6.

10. Figure S4b. Where are the controls to determine the basal levels of H2AX phosphorylation and the specificity of the effect (e.g. does it happens when adding another exogenous RNA at similar concentrations)?

11. Overall figure legends must be expanded to allow readers to understand the experiments (see above). For example, where is derived the fluorescence measured in Fig 5c derived from? How is this related to necrosis?

Minor points

Page 5. Why is the data from DNase treatment not shown? I think this is very important data and should be presented to back up the conclusions.

Figure 1. Not sure LTCM, BTCM and ETCM are defined in the text or figure legend.

Figure 2a. What is the label of the Y axis in the barplot?

Figure S2c. The co-localisation experiment is not very clear and to me it seems that Zc3h12d may co-localise partially with almost every marker showed. A more refined analysis using monitoring the fluorescence distribution profile of each fluorophore should be done to draw any conclusion.

Page 7. Do authors refer to FITC-labelled RNA with or without cap and poly(A) tail? Otherwise I do not understand what authors meant by 'its cap and poly(A) adducts'. I suggest to be very clear here.

Figure 4e-f. Wild type or wt instead of Wild.

Reviewer #2 (Remarks to the Author):

In their manuscript “Extracellular mRNA transported to the nucleus exerts translation-independent function”, Tomita et al describe a novel function for the protein ZC3H12D as a mediator RNA uptake from the extracellular space. They also provide evidence that ZC3H12D-mediated uptake of non-vesicular, extracellular IL1B mRNA leads to cellular stress, and, in the case of a specific NK cell population, their activation and increased anti-tumoral activity.

Although ZC3H12D is known to be an RNA binding protein, up to now its function has been assumed to be similar to that of ZC3H12A or “Regnase”, which binds and degrades the mRNA of specific proinflammatory cytokines. Thus, the RNA uptake function described by Tomita and colleagues is truly novel and surprising. Moreover, the described non-transcriptional function of IL1B mRNA in NK cells is also novel and unexpected.

Altogether, this study represents a significant advance for the field and will be of great interest to immunologists and cell biologists. However, precisely due to the really surprising nature of its findings, this reviewer also thinks that several control experiments are necessary before the manuscript is suitable for publication. In general, these controls include (i) investigating whether other mRNAs are also bound and internalized by ZC3H12D, in particular those found in the array in Figure 1 / Table S2, (ii) using β -actin or another non-ZC3H12D-binding RNA as a control for experiments, (iii) including human and other murine cell lines for RNA uptake, cellular stress, etc. The specific experiments are detailed below.

Figure 1

Major points:

- Figure 1a: According to array data in biogps (<http://biogps.org/#goto=genereport&id=340152>), ZC3H12D should be broadly expressed in PBMCs. However, it is unclear if all of these cells really express ZC3H12D protein and internalize ZC3H12D upon exposure to tumor supernatant. Could the authors please co-stain for T-cells, B-cells, NK-cells and Monocytes in this experiment? This would provide valuable information on the role of ZC3H12D in these cell types.
- The authors then switch to the use of RAW macrophages. Since murine macrophages also express ZC3H12D, these data are valuable, but do not necessarily reflect the situation in PBMC. Could the authors also include another cell line but perhaps more representative for blood immune cells, e.g. a murine T-cell or B-cell line, if necessary with overexpression of ZC3H12D?
- Since data with human tumor cells and supernatants are also used in the figure, human PBMC and/or using a human leukocytic cell line, e.g. THP-1, Daudi, NK-92 cells, should be included to visualize ZC3H12D internalization. If no antibody can be found, this could be done using overexpressed tagged protein in a one of the cell lines.
- Please include the other genes from the gene expression array/ Table S2 (immunoglobulin kappa variable (Igkv), resistin like gamma (Retnlg), matrix metalloproteinase 8 (MMP8)) in the experiments in Fig 1h-j. It would thus greatly improve the scope of this study, if other RNAs were investigated (see section on the discussion).

-

Minor points:

- What cells are shown in 1c? "Some cells from the spleen" is rather vague.
- Could you include a graphical schema of the experiment in 1g? This would greatly help the reader.
- Fig 1j would fit better in Figure 2.

Figure 2

Major points:

- As above, the experiment in Fig. 2a should also be performed with 1-2 further cell lines (murine blood cells, human cells)
- The FCS experiment in Fig. 2c should also be performed with a non-binding RNA such as β -actin as a control to make the data easier to interpret. Doesn't the 3'UTR of IL1B also interact with ZCH12D?

Minor points:

- Please calculate the colocalization in Fig. 2a using Pearson, Manders or a similar approach.

Figure 3:

Major points:

- As above, please include a human cell line (preferably THP1 and/or NK-92) in the experiments for Fig. 3c and 3d.
- Please also use a control RNA such as β -actin for the experiments in Fig. 3c and 3d.

Minor points:

- The bar graph in Fig 3d is slightly distorted.

Figure 4:

Minor points:

- Please use the space above Fig. 4b to label each the bp of each segment used for EMSA. Despite the detailed information in the supplementary figures, putting the bp range in the figure would greatly help the reader. If the reader doesn't realize that the probes overlap, it is hard to understand why probe 5 was chosen for further investigation.
- Fig. 4f would fit better in Figure 5.

Figure 5/6:

Major points:

It is undoubtedly interesting that IL1B mRNA induces H2AX phosphorylation in a ZC3H12D-dependent manner, but it does not necessarily mean that cell-intrinsic cell stress leads to NK-cell activation and IFN γ induction --in particular, because H2AX phosphorylation is indicative of DNA damage and often followed by apoptosis (rather than increased survival) in other cell types.

Please address the following questions:

- Does the observed response to IL1B mRNA/H2AX phosphorylation specific to NK cells?
 - oCan other DNA-damaging agents (e.g. camptothecin) provoke a similar response (activation, IFN γ , increased cytotoxicity) in NK cells?
 - oDo other cell types (RAW macrophages, etc) react differently to ZCH12D-mediated IL1B mRNA uptake? Is there H2AX phosphorylation? Do cells become apoptotic?
- As NK cells can be activated by interaction with “stressed cells”, is the response really cell intrinsic? This can be tested by incubating other cells, such as IL1 β RNA-stimulated WT and Zc3h12d $^{-/-}$ BMDM, with NK cells in an in vitro killing assay. Alternatively, one could overexpress ZC3H12D in a target cell line.
- Are the genes upregulated in ZC+ RAW in Fig 5e and 5f also upregulated in B220+CD11c+NK1.1+ cells?
 - oHow were the genes in Table S3 picked? Are they part of the same microarray as in figure 1?
- In the assays in Fig 6b and 6c, it would be important to include a control RNA (e.g. β -actin). Although the authors have thought to cap and 2’O-methylate the IVT mRNA, unmodified RNA could activate RIG-I in NK cells, which also leads to their activation. (In Fig. 6a this control was performed.)

Minor points:

- The necrosis assay in 5c should be explained a little. It is somewhat unclear what exactly was measured.

Figure 7:

Major points:

- It is interesting that IL1 β mRNA induces IFN γ in the CD56 $^{\dim}$, but not the CD56 $^{\text{bright}}$, NK cell population. Please compare IL1B mRNA uptake in these cell populations.
- If the uptake is still observed for CD56 $^{\text{bright}}$ NK cells, does IL1B mRNA uptake lead to the upregulation of other NK cell activation markers on CD56 $^{\text{bright}}$ NK cells (e.g. NKG2D, NKp46)? What about killing activity?

Discussion: Several important points are currently missing in the discussion.

- Please place your data into the context of what has been previously published about ZC3H12D, i.e. you do not observe RNase activity, you do not observe “Regnase-like” activity.
- Which human NK cell population is closer to B220+CD11c+NK1.1+? According to Blasius et al, 2007. These cells should be more similar to the CD56 $^{\text{bright}}$ human population, yet this doesn’t fit with your observations. This should be discussed.
- Do you think that ZC3H12D is also involved in the uptake of other RNAs? ZC3H12D is evolutionarily highly conserved, with putative orthologs found even in protozoa(<https://www.uniprot.org/uniprot/?query=taxonomy:5811%20zc3h12d>). In contrast, IL1B is only found in vertebrates. Thus, it seems highly unlikely that ZC3H12D only acts to transport IL1B mRNA.
- How do these extravesicular RNAs survive in the serum?

References

Blasius AL, Barchet W, Cella M, Colonna M. Development and function of murine B220+CD11c+NK1.1+ cells identify them as a subset of NK cells. *J. Exp. Med.* 2007 Oct 29;204(11):2561–8.

Reviewer #1 (Remarks to the Author):

#Q1: Authors report here a very interesting story that if is true will be paradigmatic for extracellular RNA function. However, with the data presented and the controls missed, I am not sure that all the central claims of this work are well supported. It is often the case that figures lack critical controls or seem incomplete (see below). I have also missed the use of a marker to delimit the cytoplasm in nucleus/cytoplasm localisation studies (e.g. Phalloidin) as the cytoplasm of the cells used appears to be small. Anyway, I think the 'killing' experiment is still missing. It would be to perform the experiments with an IL1B mRNA at physiological concentration including one or more stop codons to avoid the production of IL1B protein. This, with additional controls to show that the toll-like receptor pathway is not the cause of the observed effects (and I think this possibility is likely), would give a lot of value to the paper.

#A1: We appreciate the reviewer's comments; they were very thoughtful and constructive. We have responded to them and believe we have provided more concrete evidence, especially concerning the translation-independent role of nex-mRNA, in the revised manuscript.

Major points

#Q2: 1. Page 4-5. I guess it would be good to highlight that the experiment treating cells from spleen with TCM has as a goal to show that cells that are not within the tumour environment can respond to TCM treatment. Why authors did not compared cells from wt vs ZC3H12D KO mice?

#A2: We added a figure of splenic leukocytes derived from wild-type and Zc3h12d^{-/-} mice in Fig1d. We also added the data pertaining to how DNase-treated TCM and purified RNA from TCM stimulated the relocation of ZC3H12D protein from the cell surface. We modified Fig1d and sentences in the revised text as shown below. (page 5,

line 2-7)

Strikingly, pretreatment of TCM with RNase (TCM+RNase) did not affect the localization pattern of ZC3H12D (wt panel in Fig. 1d). The addition of RNA isolated from TCM resulted in the same phenomenon (wt panel, RNA-TCM column in Fig. 1d). These data suggest that RNA drives the translocation of ZC3H12D. On the contrary, this was not observed when we used splenic leukocytes *from Zc3h12d*^{-/-} mice (Fig 1d).

#Q3: 2. I don't understand the reasoning behind using spleen gene expression profile to identify the potential regulatory nex-mRNAs. Wouldn't make sense to analyse the supernatant of cultured tumours (TCM) after purification of extra vesicle RNA? Is really the analysis used adequate to identify nex-mRNA by transcriptomic profiling? Maybe I have missed something here. Moreover, I don't think I fully understand the reasoning behind the selection of IL1B mRNA as from my point of view is not fully justified here. Weren't other RNAs as upregulated as IL1B mRNA (based on Fig S4)? Was this a lucky guess or there was a reason to think that IL1B mRNA was going to be a) extracellular and extra vesicle and b) would produce a phenotype. As I said, I may have missed something here and if so, apologies for this.

#A3: We apologize for the confusion. We did not show whether the lung-conditioned media from lungs cultured with TCM (Lung-TCM) contained *IL1 β -mRNA* and *β actin-mRNA*. We added the new microarray data (GSE161219) in a revised Table S2. High amounts of non-vesicular *IL1 β -mRNA* and *β actin-mRNA* were detected in the Lung-TCM, and both genes were ranked in the top 20 genes in the microarray dataset. We obtained candidates including *IL1 β -mRNA* and *β actin-mRNA* from the Lung-TCM data. These genes should be further screened to compare wild-type, and *Zc3h12d* knockout mice since the expression level of ZC3H12D-related genes were affected by *Zc3h12d* knockout. Preparation of nex-mRNA from Lung-TCM requires many mice at once. Due to our animal facility's limitations, we were unable to prepare samples from *Zc3h12d* knockout mice. In addition, ZC3H12D expression was prominent in the spleen. We considered that the spleen is more sensitive than other tissues in terms of the

ZC3H12D knockout effect. Thus, we conducted a second screening using the microarray data shown in the revised Table S3 (Table S2 in the first submitted paper) generated by wild-type and *Zc3h12d* knockout mice. Based on the data in Table S3, we decided to add an *immunoglobulin kappa variable (Igkv)*, *resistin like gamma (Retnlg)*, *matrix metalloproteinase 8 (Mmp8)*, *βactin*, and *gapdh* data analyses to Fig. 1. This criticism was also mentioned by reviewer 2. We used *βactin* as a negative biological control in this paper. We added sentences in the revised text shown below. (page 5, line 16-28)

Based on our data that TCM induced RNA-depending response of ZC3H12D location (Fig. 1d), we hypothesized that some specific exRNA²⁴ bind to ZC3H12D. Because exRNA should be exposed to the ZC3H12d protein on leukocytes, we tried to examine which non-vesicular exRNA (nex-RNA) was increased in lung microenvironment that was stimulated by a tumor. We carried out a microarray analysis on the lungEC-TCM. The top 20 ranked genes are listed in the table (Table S2). We then searched for mRNA differences in expression between wild-type and *Zc3h12d*^{-/-} mouse spleens. Spleens were chosen because they had the highest ZC3H12D expression compared to other organs tested in this study. The rationale was that ZC3H12D depletion effects might be seen by comparing wild-type and *Zc3h12d*^{-/-} spleen data. This comparison revealed that *immunoglobulin kappa variable (Igkv)*, *resistin like gamma (Retnlg)*, *interleukin 1 beta (Il1β)*, and *matrix metalloproteinase 8 (Mmp8)* were upregulated in the *Zc3h12d*^{-/-} sample (Table S3). Among the upregulated genes, we focused on *IL1β-mRNA* because this gene exhibited relatively high expression of the nex-RNA derived from TCM-stimulated lung cells (Table S2). In addition, we chose *βactin* as a negative control in our study because it was abundantly found in the TCM-stimulated lungs (Table S2), and its expression levels did not change between wild-type and *Zc3h12d*^{-/-} mice (Table S3).

#Q4: 3. Page 6. I am not familiar with the extra-exosomal sample preparation used here for the RNA analysis by RT-qPCR. After reading the mat and met I am still confused as I am not familiar with the kits. Maybe some sentences to clarify the purpose of each step would help. Do authors control for extra-exosomal RNA using RNase treatment (that should digest

unprotected RNA)?

#A4: Exosomes can be fractionated by size exclusion chromatography, ultracentrifugation, magnetic bead pull-down, and polymer precipitation. ExoQuick precipitation procedure (Methods Mol Biol, 728, 235-246, 2011) is one of the most popular methods to isolate exosomes from biological samples. ExoQuick is a synthetic polymer and relatively easy to handle. In addition, ExoQuick enables us to completely remove exosomes from the condition media without interfering with the extra-exosomal RNAs. Size exclusion column chromatography and ultracentrifugation dilute sample too much. Magnetic beads have a non-specific binding ability to absorb extra-exosomal RNA. These treatments cause loss of extra-exosomal RNA in the samples and affect our data quality. We concluded that the ExoQuick-TC reagent was suitable for our experiments. After mixing the reagent with a conditioned medium, it was kept stable at 4 degrees Celsius. The polymer deprives water molecules around the exosome to reduce their solubility, enabling us to precipitate the exosomes by table-top centrifuge. We added the method scheme in Fig. 1g and a brief explanation as below in the material method section. (page25, line 22-25) We also presented RNase treated TCM (TCM+RNA) data in Fig. 1d to show effects of the digestion of extra-exosomal RNA.

Exosomes can be fractionated by size exclusion chromatography, ultracentrifugation, magnetic bead pull-down, and polymer precipitation (Methods Mol Biol, 728, 235-246, 2011). Among them, ExoQuick-TC (System Biosciences), a polymer used to precipitate exosomes, is the best method to eliminate exosomes from conditioned media.

#Q5: 4. Page 6 / Figure 2a. While authors discuss an interaction between ZC3H12D and IL1B mRNA, the microscopy shows a clear exclusion of the fluorescence derived from each molecule. Since the microscopy used is expected to have low resolution (200-300 μm), if both molecules do interact, they would be expected to be present at (at least partially) overlapping areas instead of juxtaposed. Moreover, I don't understand why other FITC-labelled RNAs weren't used to show specificity and why the NoCM control is not shown in the microscopy above.

#A5: We included a NoCM control- and TCM- induced ZC3H12D co-localization with FITC-labeled *IL1 β -mRNA* in Fig. 2a and Fig. S2a. In these images, Phalloidin was used as a marker. To quantify the co-localization signals of FITC-labeled *IL1 β -mRNA* and ZC3H12D, we calculated Pearson's R values and described them in the revised text. This criticism was also mentioned by reviewer 2. In addition, we showed clear co-localization data without Phalloidin staining (For reviewer data 1). We found that the signal became obscure after the membrane perforation process, required for Phalloidin staining. We added the sentence below to the text. (page6, line 27-29)

Thirty minutes after TCM stimulation, the co-localization signals in ZC⁺RAW cells were more prominent (Pearson's R value is 0.47 ± 0.04 , n=3) than the stimulated control, NoCM (arrow in Fig. 2a and Fig. S2a).

#Q6: 5. Fig. 2b. How is copy number estimated? Why other (control) RNAs aren't tested? Ideally this immunoprecipitation should be combined with RNAseq of the eluates to provide an overview of the RNAs bound by ZC3H12D and show that IL1B mRNA is a dominant target. Otherwise, it is difficult to justify why focusing on IL1B mRNA and not others. It is also difficult to identify positive and negative RNA controls to use in downstream experiments.

#A6: The reason for selecting *IL1 β -mRNA* as the target gene and other control genes such as *β actin*, *gapdh*, *retnlg*, *mmp8*, and *Igkv* is described above. Estimations of copy number and physiological concentrations of RNA are discussed below.

We quantified *IL1 β -mRNA* in various TCMs or Lung-TCMs using a real-time PCR. Data for the standard curves were prepared using diluted *IL1 β -cDNA*. Copy number of the standard sample was calculated based on the absorbance at 260 nm. Real-time PCR gave us cycle threshold (Ct) values for *IL1 β -mRNA* of the various TCMs or Lung-TCMs in Fig. 1. These Ct values were then converted to copy numbers by using the standard curve. To further convert copy number into concentration, we used the

molecular weight of full-length *IL1 β -mRNA* as 300×1500 (nts) = 4.5×10^5 (g). Thus, 1 ng of full-length *IL1 β -mRNA* corresponds to 1.3×10^9 copies of the molecule. While this is not accurate, we considered it an acceptable approximation. According to this estimate, conditioned media contained about 1 pg/ml of extra-exosome RNA. Based on Fig. 2b, the amount of *IL1 β -mRNA* bound to ZC3H12D⁺ cells was almost 200~300 copies after the application of Lung-TCM originally containing 1.3×10^6 copies of *IL1 β -mRNA*. Given the cell numbers, sample degradation in the media, and sample loss during the immunoprecipitation process, roughly 0.1%-1% of *IL1 β -mRNA* was detected in ZC3H12D⁺ cells within TCMs or Lung-TCMs. In the experiments, we observed clear biological responses such as migration and IFN γ induction of ZC3H12D⁺ cells by 1~10 ng/ml of *IL1 β -mRNA*. In addition, confocal microscopy systems required at least 10 ng/ml of FITC-labeled *IL1 β -mRNA* to detect signals in the nucleus. Moreover, it was reported that 1,000-5,000 ng/ml of B-DNA and poly (I:C) were used as stimulators on nucleic acid-mediated innate immune responses in vitro (PNAS 108: 11542-11547, 2011). We also showed that passing TCM through a ZC3H12D protein column did not induce the migration activity in Fig S7a. Taken together, although it would be difficult to estimate the physiological concentration of *IL1 β -mRNA*, somewhere between 1 pg/ml ~ 1 ng/ml of *IL1 β -mRNA* might be expressed when tumors influence the tissues. In this revision, we carried out most experiments using minimal concentrations (10 ng/ml = 22 pM) of RNA, as the reviewer suggested.

#Q7: Figure legends of panels d-e are not sufficiently elaborated to understand what these plots show (at least for people not familiar with them). At a glance, I can see some coherency regarding the Y axis, but a large data distribution in the X axis. Is this what authors expected?

#A7: We re-analyzed data and rebuilt the figures, newly named as Fig 2c-f. These figures show FCS curve shift by the addition of RNA. This indicates a direct interaction between ZC3h12D protein and RNA. To simplify the logic of this study, only FCS curves and bar graphs for the protein-RNA complex ratio were shown.

#Q8: 6. Fig. 3a. A control with unlabelled IL1B mRNA is missing. The signal observed is very close to noise levels, so this control is critical to differentiate between signal and noise. Moreover, the experiments in Figure 3 are generated with large amounts of exogenous IL1B mRNA. Authors should prove that in physiological TCM IL1B mRNA concentrations (i.e. the one found in TCM), the RNA would be detected in the nucleus.

#A8: This is an important point. We repeated all the data acquisitions using non-labeled *IL1 β -mRNA*, and these data were used as basal signals. As described above, we used 10 ng/ml (22 pM) of *IL1 β -mRNA* instead of 100 ng/ml. As shown in Fig. 3, the signal intensities remained at a low level when 10 ng/ml of *IL1 β -mRNA* was used, but at the same time, the difference between the labeled and non-labeled *IL1 β -mRNA* became significant. We also added *IL1 β -stop-mRNA* data as suggested by the reviewer. We added the sentences below to the revised text. (page 7, line 27-31)

To evaluate whether exogenous *IL1 β -mRNA* was translated to functionally modify the cell, we prepared *IL1 β -mRNA* with nonsense mutations (*IL1 β -stop-mRNA*). This RNA has three stop codon mutations in the protein coding sequence so that it does not generate IL1 β protein. Interestingly the *IL1 β -stop-mRNA* clearly transported to the nucleus (Fig 3a right).

#Q9: 7. Figure 3c. Not sure what is the Wt panel as authors only show *Zc3h12d* +/- and -/-. Without the +/+ control this experiment is inconclusive and difficult to evaluate. All the experiments in figure 3 should include an unlabelled RNA control to define the autofluorescence level. Without this, it is very difficult to assess whether the observed fluorescence is signal or autofluorescence.

#A9: We carried out the uptake of non-labeled *IL1 β -mRNA*, FITC-labeled- *IL1 β -mRNA*, and *IL1 β -stop-mRNA* in B220⁺CD11c⁺NK1.1⁺ cells derived from TCM-stimulated wild-type and *Zc3h12d*^{-/-} mice. This data is shown in Fig 3c.

#Q10: 8. Figure 4. Can authors show what is the affinity of Zc3h12d for the target RNA in the EMSA assay?

#A10: Several EMSA with different protein concentrations must be done to determine the affinity of the ZC3H12D protein for the target RNA. Normally, researchers depict a correlation graph showing protein concentration vs. binding probe ratio, and the protein-probe affinity is calculated according to the curve fitting results. In this study, EMSA was conducted by one concentration only; it is not possible to draw any conclusion. In Fig. 4b, probe 5 displays a shifted band slightly more intense than the unshifted band. This implies that protein concentration in this assay was close to its Kd value, allowing us to make a very rough estimation of 10 nM-100 nM. To make a solid scientific statement about the affinity of the ZC3H12D-RNA target, we need to have more data. We want to let you know that our biochemical project to analyze the ZC3H12D-RNA target interaction in detail is ongoing.

#Q11: In Figure 4e, without a splitting of the channels, the inclusion of a negative RNA control, and a paralleled Zc3h12d KO line analysis, it is not possible to fully interpret and draw solid conclusions from this microscopy experiment.

#A11: We compared the uptake of FITC-labeled-*IL1 β -mRNA* in B220⁺CD11c⁺NK1.1⁺ cells derived from TCM-stimulated wild-type, *Zc3h12d*^{-/-}, and Regnase-1^{-/-} mice. In this experiment, the same littermate pairs were used. We used *β actin-mRNA* as a negative control. The data is shown in Fig. 5a and Fig. S5b.

#Q12: 9. Figure 5. If the effect is independent of translation, I would expect to see the same effect if one or more stop codons are inserted into the IL1B mRNA to avoid the production of the protein. This would be the only way to show that the effect is translation-independent although it cannot be excluded the possibility that IL1B mRNA interaction with membrane

proteins triggers a signalling cascade through, for example, RNA-binding toll-like receptors. Controls showing that toll-like receptor pathway is not activated should be shown. Unfortunately, the Cap/poly(A) – is a good control but not sufficient to rule translation and toll-like receptor roles in the observed effects. 1) An mRNA lacking cap and poly(A) can be translated if added in large quantities. 2) An mRNA lacking cap and poly(A) will exhibit lower stability as it is susceptible to exonucleases. These two factors can explain why authors see the effects when adding large amounts of cap and poly(A) less RNA. Controls using RNAs with near identical sequence but that cannot be translated are thus essential here. Also showing that other RNAs do not produce this effect when used at the very same (or even higher) concentration, would be critical to rule out RNA sensing and triggering of the antiviral programme. This criticism is applicable to Fig 6.

#A12: Thank you for this thoughtful comment. We repeated the experiments using *IL1 β -mRNA*, *IL1 β -stop-mRNA*, and the controls were *β actin-mRNA* and *gapdh-mRNA* with cap and poly(A) shown in Fig. 3, Fig. 5, and Fig. 6. The *IL1 β -stop-mRNA* data supported that the effect of *IL1 β -mRNA* was translation-independent. We added the following sentences in the discussion. (page 16, line 21-23)

Mutant mRNA with multiple stop codon mutations also elicited the same phenomenon, indicating that the exotic mRNA functions in a translation-independent manner.

#Q13: 10. Figure S4b. Where are the controls to determine the basal levels of H2AX phosphorylation and the specificity of the effect (e.g. does it happen when adding another exogenous RNA at similar concentrations?)

#A13: We carried out the H2AX phosphorylation immunostaining using *β actin-mRNA* and *IL1 β -mRNA*. Data are shown in Fig. 5b and Fig. S5d.

#Q14: 11. Overall figure legends must be expanded to allow readers to

understand the experiments (see above). For example, where is derived the fluorescence measured in Fig 5c derived from? How is this related to necrosis?

#A:14 We added explanations in the legends. We added the following sentence to the legend for Fig. 5c. (page 41, line 1-2)

Real-time tracing of fluorescein-DNA dye indicates loss of membrane integrity resulting in late-stage apoptosis.

Minor points

#Q15: Page 5. Why is the data from DNase treatment not shown? I think this is very important data and should be presented to back up the conclusions.

#A15: We added the DNase data to Fig. 1d.

#Q16: Figure 1. Not sure LTCM, BTCM and ETCM are defined in the text or figure legend.

#A16: We explained them in the legend for Fig. 1 as below. (page 38, line 17-18)

TCMs, such as LLC-TCM, E0771-TCM, and B16-TCM, are presented as LTCM, ETCM, and BTCM, respectively.

#Q17: Figure 2a. What is the label of the Y axis in the barplot?

#A17: Instead of Fig. 2a, we showed the result of the Pearson's R value to evaluate the co-localization signals using Image J/Fiji (coloc 2) in the text, and the method was

described in the methods section. (page31 line 11)

#Q18: Figure S2c. The co-localisation experiment is not very clear and to me it seems that Zc3h12d may co-localise partially with almost every marker showed. A more refined analysis using monitoring the fluorescence distribution profile of each fluorophore should be done to draw any conclusion.

#A18: We calculated the co-localization signal of ZC3H12D and nuclear body markers to find out that the ZC3H12D signal was relatively high in nuclear speckles. (page 7, line 31- page 8, line 2) The quantitative data is shown in (Fig. S2d).

#Q19: Page 7. Do authors refer to FITC-labelled RNA with or without cap and poly(A) tail? Otherwise I do not understand what authors meant by 'its cap and poly(A) adducts'. I suggest to be very clear here.

#A19: We described RNA with a cap and poly(A) tail as RNA Cap(+)Poly(A) in all figures.

#Q20: Figure 4e-f. Wild type or wt instead of Wild.

#A20: We used 'wt' as reviewer suggested.

Reviewer #2 (Remarks to the Author):

#Q21: In their manuscript “Extracellular mRNA transported to the nucleus exerts translation-independent function“, Tomita et al describe a novel function for the protein ZC3H12D as a mediator RNA uptake from the extracellular space. They also provide evidence that ZC3H12D-mediated uptake of non-vesicular, extracellular IL1B mRNA leads to cellular stress, and, in the case of a specific NK cell population, their activation and increased anti-tumoral activity.

Although ZC3H12D is known to be an RNA binding protein, up to now its function has been assumed to be similar to that of ZC3H12A or “Regnase”, which binds and degrades the mRNA of specific proinflammatory cytokines. Thus, the RNA uptake function described by Tomita and colleagues is truly novel and surprising. Moreover, the described non-transcriptional function of IL1B mRNA in NK cells is also novel and unexpected.

Altogether, this study represents a significant advance for the field and will be of great interest to immunologists and cell biologists. However, precisely due to the really surprising nature of its findings, this reviewer also thinks that several control experiments are necessary before the manuscript is suitable for publication. In general, these controls include (i) investigating whether other mRNAs are also bound and internalized by ZC3H12D, in particular those found in the array in Figure 1 / Table S2, (ii) using β -actin or another non-ZC3H12D-binding RNA as a control for experiments, (iii) including human and other murine cell lines for RNA uptake, cellular stress, etc. The specific experiments are detailed below.

#A21: We appreciate your valuable comments. We repeated the experiments using β actin and genes found in Table S2 (Table S3 in the revised manuscript) as suggested by the reviewer. Our FCS revealed that ZC3H12D has a non-specific binding site for RNA, but EMSA and other biological assays clarified that ZC3H12D recognizes the RNA sequence. Thus, we used the above mentioned RNAs, including β actin, as a

negative control. In addition, we included a new human cell line, THP-1, as recommended. In the revised manuscript, modified or newly added parts are highlighted in red. The responses are shown one by one below.

Figure 1

Major points:

#Q22: -Figure 1a: According to array data in biogps (<http://biogps.org/#goto=genereport&id=340152>), ZC3H12D should be broadly expressed in PBMCs. However, it is unclear if all of these cells really express ZC3H12D protein and internalize ZC3H12D upon exposure to tumor supernatant. Could the authors please co-stain for T-cells, B-cells, NK-cells and Monocytes in this experiment? This would provide valuable information on the role of ZC3H12D in these cell types.

#A22: Thank you for the comments. We found that ZC3H12D was on the cell surface of T cells and B cells in tumor-conditions. Their populations were lower than that of NK cells (For reviewer data 2). Because further investigations are needed to decipher the cell specific biological roles of ZC3H12D, we are running a new project.

#Q23: -The authors then switch to the use of RAW macrophages. Since murine macrophages also express ZC3H12D, these data are valuable, but do not necessarily reflect the situation in PBMC. Could the authors also include another cell line but perhaps more representative for blood immune cells, e.g. a murine T-cell or B-cell line, if necessary with overexpression of ZC3H12D?

-Since data with human tumor cells and supernatants are also used in the figure, human PBMC and/or using a human leukocytic cell line, e.g. THP-1, Daudi, NK-92 cells, should be included to visualize ZC3H12D internalization. If no antibody can be found, this could be done using

overexpressed tagged protein in a one of the cell lines.

#A23: We examined THP-1 cells to find out that a small number of cells expressed ZC3H12D on the cell surface, and their intracellular relocations upon stimulation by MDAMB231-derived TCM were observed. However, the responses were weaker than NK cells. This data was described in the text and Fig. S3a as below. In the revised manuscript, *IL1 β -stop-mRNA* having 3 stop codon mutations in the coding sequence was tested to clarify that the biological functions of *IL1 β -mRNA* analyzed in this study were translation-independent. This point was also mentioned by reviewer 1. (page 8, line 2-8)

In addition, our flow cytometric analysis of THP-1 cells revealed that ZC3H12D cell surface expression was observed in 0.3% of the cells with conditioned media (CM) stimulation. In contrast the population decreased to 0.1% in the presence of TCM stimulation. This decrease indicates TCM stimulated intracellular internalization of ZC3H12D from the cell surface. The ZC3H12D⁺THP1 cells incorporated *IL1 β -mRNA* and *IL1 β -stop-mRNA* into the nucleus, although the signals were weaker than in NK cells (Fig. S3a).

#Q24: -Please include the other genes from the gene expression array/ Table S2 (immunoglobulin kappa variable (Igkv), resistin like gamma (Retnlg), matrix metalloproteinase 8 (MMP8)) in the experiments in Fig 1h-j. It would thus greatly improve the scope of this study, if other RNAs were investigated (see section on the discussion).

#A24: Thank you for the important comment. We ran qPCR reactions for *immunoglobulin kappa variable (Igkv)*, *resistin like gamma (retnlg)*, *matrix metalloproteinase 8 (mmp8)*, *gapdh*, and *β actin* and added data in Figs. 1i and 1j.

Minor points:

#Q25: -What cells are shown in 1c? “Some cells from the spleen” is rather vague.

-Could you include a graphical schema of the experiment in 1g? This would greatly help the reader.

-Fig 1j would fit better in Figure 2.

#A25: We showed splenic leukocytes after eliminating the red blood cells in Fig 1c, then we narrowed it down to NK cells because we had reported that those cells worked with anti-metastatic ability in the lungs. In the figure and text, we specified them as splenic leukocytes. We also renewed the data to show the relocation of ZC3H12D on splenic leukocytes derived from wild-type and *Zc3h12d*^{-/-} mice in tumor-conditions in Fig. 1d, as reviewer 1 requested.

We added graphical schema to Fig. 1f (Fig. 1g in the first submission paper).

In the viewpoint of physical interactions between ZC3H12D and *IL1 β -mRNA*, it would be displayed in Fig. 2. In this case, because we would like to emphasize the existence of nex-mRNA in biological samples, we decided Fig 1j data will stay in Fig. 1.

Figure 2

Major points:

#Q26: -As above, the experiment in Fig. 2a should also be performed with 1-2 further cell lines (murine blood cells, human cells)

#A26: We added the ZC3H12D⁺THP-1 cell data. The cells were purified using a cell sorter and anti-ZC3H12D antibody. While we tried to establish ZC3H12D-overexpressing cells in murine and human cells, only 2 cell lines, murine RAW and human 786-O, have been established and used in this study. We observed the relocation of ZC3H12D protein from the cell surface to inside the cell by stimulation with *IL1 β -mRNA* but not *β actin* in THP-1 cells (Fig. S2b). Incorporated *IL1 β -mRNA* was transferred into the nucleus (Fig S3a). We also calculated the co-localization signals between the ZC3H12D protein and FITC-labeled *IL1 β -mRNA* to show Pearson's R value in the text. (page 6, line 23-32)

We next examined whether nex-mRNA could bind to ZC3H12D on the cell surface. As

ZC3H12D is expressed in murine RAW 264.7 macrophage cells ^{21, 22}, and human THP-1 cells (Mol. Cancer Res7, 880-889, 2009), we used them. To visualize the interaction between *IL1 β -mRNA* and ZC3H12D protein, we set up a mouse ZC3H12D-overexpressing RAW cell line (ZC⁺RAW) and applied FITC-labeled *IL1 β -mRNA*. Thirty minutes after TCM stimulation, the co-localization signals in the ZC⁺RAW cells were more prominent (Pearson's R value is 0.47 ± 0.04 , n=3) than cells stimulated by control NoCM (arrow in Fig. 2a and Fig. S2a). In ZC3H12D⁺THP-1 cells, isolated using a cell sorter and an anti-ZC3H12D antibody, similar co-localization signals were observed after applying *IL1 β -mRNA-FITC* (Fig. S2b, Pearson's R value is 0.62 ± 0.08 , n=3).

#Q27: -The FCS experiment in Fig. 2c should also be performed with a non-binding RNA such as β -actin as a control to make the data easier to interpret. Doesn't the 3'UTR of IL1B also interact with ZCH12D?

#A27: FCS has been used to demonstrate direct protein-RNA interactions in vitro. One reference was added in the text to support this statement. We performed FCS measurements using *β actin-mRNA* and *IL1 β -mRNA* to find out both RNA bound ZC3H12D equally. The 3'UTR *IL1 β -mRNA* also bound ZC3H12D. The results revealed that ZC3H12D has a non-sequence-specific binding site. On the other hand, EMSA clearly showed that ZC3H12D has a sequence-specific binding site, other bioassay data supported this conclusion. In EMSA, RNA with the ARE sequence remained on the ZC3H12D protein despite the presence of 100-fold amounts of competitors. FCS shows a direct interaction between two purified objects in solution but does not distinguish specific binding sites from non-specific binding sites and we could not conduct competitor experiments in FCS. The sequence-specific binding was emphasized in the biochemical assays because FITC-labeled- *IL1 β -mRNA* but not *β actin-mRNA* seemed to bind to ZC3H12D on the cell membrane (Fig2a), implying that physiological binding structures play important roles. Our new project is deciphering detailed interactions between ZC3H12D and various RNAs, including *β actin-mRNA* and *IL1 β -mRNA* and their fragments, using FCS and other spectroscopic techniques. It is not surprising to see non-specific binding and sequence-specific sites in the same protein (for example, p53). A non-specific binding site may have a different role from a sequence-specific binding

site. We modified sentences in the text as shown below. (page 7, line 4-9) (page 7, line 13-17)

The data showed that the ZC3H12D protein enriched *IL1 β -mRNA* in the ZC+RAW cells (Fig. 2b). On the contrary, the *gapdh- β actin- mRNAs* control showed little enrichment in the condition (Fig 2b). We investigated direct interactions between the ZC3H12D protein and *IL1 β -mRNA* using fluorescence correlation spectroscopy (FCS). This technique has been used to demonstrate a direct protein-RNA interaction *in vitro* (PNAS 104, 12306-12311, 2007).

The FCS data indicates that the dissociation constant for ZC3H12D- *β actin* and *IL1 β -mRNA* were in the order of 1 nM. We also confirmed that the 3' untranslated region (3'UTR) of *IL1 β -mRNA* had direct interaction with the ZC3H12D protein (Fig. 2e and 2f). Because the 3'UTR of *IL1 β -mRNA* had a different sequence from *β actin-mRNA*, the binding might include non-specific and sequence-specific interactions.

Minor points:

#Q28: -Please calculate the colocalization in Fig. 2a using Pearson, Manders or a similar approach.

#A28: We described above.

Figure 3:

Major points:

#Q29: -As above, please include a human cell line (preferably THP1 and/or NK-92) in the experiments for Fig. 3c and 3d.

#A29: We carried out FITC-labeled mRNA uptake in ZC3H12D⁺THP-1 cells and found

that *IL1 β -mRNA* but not *β actin-mRNA* was incorporated in these cells, although the signals in the nucleus (Fig S3a) were 30% lower than RAW and NK cells (Fig. 3a, 3c).

#Q30: -Please also use a control RNA such as β -actin for the experiments in Fig. 3c and 3d.

#A30: We carried out experiments again using *β actin-mRNA* and *IL1 β -mRNA* in Figs 3c and 3d.

Minor points:

#Q31: -The bar graph in Fig 3d is slightly distorted.

#A31: We changed it to a new figure.

Figure 4:

Minor points:

#Q32: -Please use the space above Fig. 4b to label each the bp of each segment used for EMSA. Despite the detailed information in the supplementary figures, putting the bp range in the figure would greatly help the reader. If the reader doesn't realize that the probes overlap, it is hard to understand why probe 5 was chosen for further investigation.

#A32: We added nucleotide numbers at the top of Fig. 4b.

#Q33: -Fig. 4f would fit better in Figure 5.

#A33: We moved Fig. 4f to Fig. 5a and Fig. S5b. This was also mentioned by reviewer 1.

Figure 5/6:

Major points:

It is undoubtedly interesting that IL1B mRNA induces H2AX phosphorylation in a ZC3H12D-dependent manner, but it does not necessarily mean that cell-intrinsic cell stress leads to NK-cell activation and IFN γ induction --in particular, because H2AX phosphorylation is indicative of DNA damage and often followed by apoptosis (rather than increased survival) in other cell types.

Please address the following questions:

#Q34: Does the observed response to IL1B mRNA/H2AX phosphorylation specific to NK cells?

#A34: First, we re-examined the phenomenon in B220⁺CD11c⁺NK1.1⁺ NK cells derived from wild-type and *Zc3h12d*^{-/-} mice using 20 ng/ml but not 100 ng/ml mRNA. In the new Fig. 5b, *IL1 β -mRNA* but not *β actin-mRNA* and *gapdh-mRNA* induced phospho-H2AX signals in the nucleus of wild-type NK cells. ZC3H12D⁺THP-1 also increased phospho-H2AX signals after the application of *IL1 β -mRNA* (For reviewer Data 3). Then, we tested RAW cells. However, it was difficult to evaluate whether immortalized cell lines became apoptotic under this condition; we only presented the data for the reviewer.

#Q35: Can other DNA-damaging agents (e.g. camptothecin) provoke a similar response (activation, IFN γ , increased cytotoxicity) in NK cells?

#A35: We examined whether 1 μ M of Camptothecin (CPT) induced IFN γ in B220⁺CD11c⁺NK1.1⁺ NK cells during the 3 hr incubation time used for the mRNA, then checked IFN γ expression 24 hr after application. In this assay, CPT did not induce IFN γ in NK cells. We this data is shown as ‘for reviewer data 4’.

#Q36: oDo other cell types (RAW macrophages, etc) react differently to ZCH12D-mediated IL1B mRNA uptake? Is there H2AX phosphorylation? Do cells become apoptotic?

#A36: We described this above.

#Q37: -As NK cells can be activated by interaction with “stressed cells”, is the response really cell intrinsic? This can be tested by incubating other cells, such as IL1bm RNA-stimulated WT and Zc3h12d^{-/-} BMDM, with NK cells in an in vitro killing assay. Alternatively, one could overexpress ZC3H12D in a target cell line.

#A37: We examined whether the stressed cells, stimulated by *IL1 β -mRNA*, induced NK cell activation in a ZC3H12D-dependent manner. We first prepared bone marrow-derived macrophages (BMDMs) from TCM-stimulated wild-type and *Zc3h12d^{-/-}* mice and incubated them in an L-cell conditioned medium for 7 days. Next, BMDMs were stimulated with *IL1 β -mRNA* for 24 hr. After the *IL1 β -mRNA* stimulation, cells were washed to remove *IL1 β -mRNA*, and B220⁺CD11c⁺NK1.1⁺ NK cells from wild-type mice were co-incubated with the primed BMDMs for 24 hr. Finally, the co-incubated NK cells were mixed with labeled tumor cells for 24 hr to evaluate their viabilities of labeled tumor cells. Dead cells were counted using the Zombie Green Fixable Viability Kit (BioLegend). We did not observe significant differences between the two groups (wt-BMDM and *Zc3h12d^{-/-}*-BMDM). We showed graphic schema and data in Fig. 6d and added sentences in the text as shown below. Fig 6d’s legend was modified accordingly. (page 12, line 1-10)

In this assay system, to account for the effects of RNA priming on the tumor cells, the

NK cells were washed before being applied to tumor cells. Our data revealed that *IL1 β -mRNA* priming increased the tumoricidal activity of NK cells originating from wild-type mice but not from the *Zc3h12d*^{-/-} mice (Fig. 6c). Again, B220⁺CD11c⁺NK1.1⁺ NK cells from *Zc3h12d*^{+/-} mice had similar tumoricidal activity (Fig. S7b). To check if *IL1 β -mRNA*-primed macrophages enhance this activity, we co-incubated NK cells with bone marrow-derived macrophages (BMDMs) stimulated by *IL1 β -mRNA* (Fig 6d, upper). Both *IL1 β -mRNA*-primed BMDMs derived from wild-type and *Zc3h12d*^{-/-} mice did not have additive effects on the tumoricidal activity *in vitro* (Fig. 6d, lower).

#Q38: -Are the genes upregulated in ZC+ RAW in Fig 5e and 5f also upregulated in B220+CD11c+NK1.1+ cells?

#A38: We added B220⁺CD11c⁺NK1.1⁺ NK data in Fig. 5f and sentences in the text as shown below. (page11, line 9-10)

Furthermore, expressions of *Dusp1* and *IL1rn* were upregulated in B220⁺CD11c⁺NK1.1⁺ NK cells after the application of *IL1 β -mRNA* (Fig. 5f).

#Q39: oHow were the genes in Table S3 picked? Are they part of the same microarray as in figure 1?

#A39: We picked up the genes from the same array data shown in Table S3 and Table S4 (Table S2 and Table S3 in the first submitted paper) (GSE104002). Genes listed in the new Table S4 were picked up based on gene ontology (GO). First, we picked up genes classified as nucleus component by GO, because our focus was on exotic RNA functions transported into the nucleus. Then, genes were sorted by fold-change value to identify genes with a fold-change value larger than 1.5. They are part of the same microarray shown in Fig. 1.

#Q40: -In the assays in Fig 6b and 6c, it would be important to include a control RNA (e.g. β -actin). Although the authors have thought to cap and 2'O-methylate the IVT mRNA, unmodified RNA could activate RIG-I in NK cells, which also leads to their activation. (In Fig. 6a this control was performed.)

#A40: Thank you for this important comment. We carried out the experiments again with *β actin-mRNA* and *gapdh-mRNA* using cap and polyA as control RNAs (Figs. 6a-c).

Minor points:

#Q41: -The necrosis assay in 5c should be explained a little. It is somewhat unclear what exactly was measured.

#A41: We added a sentence in the legend for Fig. 5c shown below. (page 41, line 1-2)

Real-time tracing of fluorescein-DNA dye indicates loss of membrane integrity resulting in late-stage apoptosis.

Figure 7:

Major points:

#Q42: -It is interesting that IL1b mRNA induces IFN γ in the CD56^{dim}, but not the CD56^{bright}, NK cell population. Please compare IL1B mRNA uptake in these cell populations.

#A42: We found that both CD56^{bright} and CD56^{dim} NK cells demonstrated uptake RNA. The ratios for CD56^{bright} and CD56^{dim} NK cells were 38% and 50%, respectively. We described these numbers in the text as shown below. (page 14, line 2-4)

We purified the CD56^{bright} and CD56^{dim} NK cell fractions from human PBMCs and tested *hIL1β-mRNA* uptake separately to find that it was in 38% of CD56^{bright} and 50% of CD56^{dim} NK cells.

#Q43: -If the uptake is still observed for CD56^{bright} NK cells, does IL1B mRNA uptake lead to the upregulation of other NK cell activation markers on CD56^{bright} NK cells (e.g. NKG2D, NKp46)? What about killing activity?

#A43: We examined NKG2D expression in CD56^{bright} NK cells after the application of *IL1β-mRNA*. While NKG2D expression slightly increased in CD56^{bright} NK cells, it was unclear if the tumoricidal activity was enhanced *in vitro*. We added data exhibiting an increase in NKG2D expression in CD56^{bright} NK cells by *IL1β-mRNA*-priming in Fig. S8e and the text, as shown below. (page 14, line 11-13)

NK cell activation marker, NKG2D, in CD56^{bright} NK cells was slightly increased after the application of *hIL1β-mRNA* (Fig. S8e).

Discussion: Several important points are currently missing in the discussion.

We described the several issue about reviewer's comments in the discussion.

#Q44: -Please place your data into the context of what has been previously published about ZC3H12D, i.e. you do not observe RNase activity, you do not observe "Regnase-like" activity.

#A44: Our bottom line is that ZC3H12D has Regnase like activity. On the other hand, in our experiments, ZC3H12D did not degrade ARE containing RNA (data not shown). Our hypothesis is that ARE containing *IL1β*-RNA does not form a stem-loop structure like other Regnase substrates. It is assumed that in the case of ARE containing *IL1β*-RNA, although the RNA binds to ZC3H12D, RNA is not close enough to Mg²⁺ in the catalytic center.

We added sentences in the discussion section as shown below. (page 15, line 9 – page 16, line 4)

ZC3H12D belongs to the ZC3H12 family, and shares an N terminal domain (NTD), PiIT N-terminus like (PIN) domain, and zinc finger (ZF) domain with other members. Among them, ZC3H12A (Regnase-1) is the most deeply investigated. Its biochemical studies revealed that recombinant Regnase-1 (NTD-PIN-ZF) recognized the stem-loop element in the 3'-UTR of *IL-6 mRNA* (Scientific reports 6, 22324, 2016,) and degraded it in an Mg^{2+} dependent manner (Nucleic acids research, 40, 6957-6965, 2012). It has been reported that Regnase-1 and ZC3H12D regulate mRNA decay by recognizing the 3'UTR of *IL-2*, *IL-6*, *IL-10*, and *TNF α* (J. Immunol 192, 1512-1524, 2014). There is no surprise that both proteins regulated the same RNAs. Regarding the PIN-ZF domain sequence, Regnase-1 and ZC3H12D are similar to each other, and their aspartate residues requiring a cleavage reaction are entirely conserved. On the other hand, *IL-17a mRNA* degradation was regulated by ZC3H12D but not Regnase-1 (J. Immunol 192, 1512-1524, 2014). This minor difference in the enzymatic specificity is attributed to the difference in their amino acid sequences. The homology between these two proteins in the NTD domain is relatively low (- 45%). In our experiments, ZC3H12D bound but did not degrade ARE containing RNA (data are not shown), while ARE binding of Regnase-1 has not been reported. Thus, it is assumed that the NTD domain modified the biochemical functions of ZC3H12D. It is also assumed that ARE containing RNA bound to ZC3H12D was apart from Mg^{2+} sitting at the catalytic center so that it was not degraded as other stem-loop substrates. Furthermore, ZC3H12D binding to long synthetic RNA with ARE (50 nts) is much stronger than short synthetic RNA (20 nts), implying that the binding affinity of ZC3H12D is susceptible to various structural factors. Our FCS measurements unveiled that ZC3H12D has a non-specific binding site for long (<1,000 nts) RNA. Taken together, we speculate that long (<1,000 nts) RNA with ARE triggers a structural change in the ZC3H12D-RNA complex to give it biological functions. Phylogenetic relationship analysis revealed that the ZC3H12 family ancestor, nematode protein REGE-1, targets *ETS-4-mRNA* different from target genes by mammalian ZC3H12 and controls pro-inflammatory mRNA (BioEssays 39, 1700051, 2017). It is suggested that a unique substrate recognition ability was adopted in this protein to regulate RNA based on their structures (BioEssays 39, 1700051, 2017)

during branching between ZC3H12D and other Regnase family genes.

#Q45: -Which human NK cell population is closer to B220⁺CD11c⁺NK1.1⁺? According to Blasius et al, 2007. These cells should be more similar to the CD56^{bright} human population, yet this doesn't fit with your observations. This should be discussed.

#A45: We added sentences in the discussion section as shown below. (page 17, line 5-12)

CD56^{bright} NK cells are assumed to be the human counterpart of murine B220⁺CD11c⁺NK1.1⁺ cells; both are in lymphoid organs and effectively produce IFN γ (JEM, 204, 2561-2568, 2007). ZC3H12D expression levels in CD56^{Dim} and CD56^{bright} NK cells were like each other. However, *IL1 β -mRNA*-mediated IFN γ production was stronger in CD56^{Dim} than in CD56^{bright} NK cells. On the other hand, *IL1 β -mRNA* induced NKG2D in CD56^{bright} NK cells, implying that NKG2D dependent anti-tumor responsiveness was enhanced in the cells (JCI, 127, 4042-4058, 2017). Together, *IL1 β -mRNA* may induce tumoricidal activity on CD56^{Dim} and CD56^{bright} NK cells *in vivo*.

#Q46: -Do you think that ZC3H12D is also involved in the uptake of other RNAs? ZC3H12D is evolutionarily highly conserved, with putative orthologs found even in protozoa(<https://www.uniprot.org/uniprot/?query=taxonomy:5811%20zc3h12d>). In contrast, IL1B is only found in vertebrates. Thus, it seems highly unlikely that ZC3H12D only acts to transport IL1B mRNA.

#A46: We agree with the reviewer's comments. We did not find other nex-mRNAs in our assay system but based on the phylogenetic tree of the Regnase family using an expanding recognition motif; it might be possible. We included this part in the discussion section.

#Q47: -How do these extravesicular RNAs survive in the serum?

#A47: We estimate that nex-mRNA might be present with co-operative proteins as RNPs. We are working to detect those in our next project.

References

Blasius AL, Barchet W, Cella M, Colonna M. Development and function of murine B220+CD11c+NK1.1+ cells identify them as a subset of NK cells. *J. Exp. Med.* 2007 Oct 29;204(11):2561–8.

We have included this reference.

For reviewer data 1

Figure 2a-related

The merged signal of *IL-1 β -RNA-FITC* and ZC3H12D protein in ZC⁺Raw cells.

For reviewer data 2

The reduction of ZC3H12D on the cell surface in TCM-stimulating mice

For reviewer data 3

Fig. 5b-related

ZC3H12D⁺THP-1 cells

pH2AX signal in ZC3H12D⁺THP-1 cell by RNA

For reviewer data 4

CPT data

B220⁺CD11c⁺NK1.1⁺ cells were incubated with 1 μ M of camptothecin (CPT) in 1% RPMI in 3hr and washed out. Twenty four hr after stimulation, IFN γ staining was carried out.

REVIEWERS' COMMENTS

Reviewer #1 (Remarks to the Author):

The authors have done an extensive work and replied most of my queries. I think the experiments with the stop codons add great value to this work. Here my final comments:

- Figure 1d. There is some ZC3H12D in ZC3H12D KO cells. Can authors comment on this? It would be nice to have error bars and t test analysis to see if the differences are significant.
- RT-qPCR experiments. Can authors add error bars and statistics to all RT-qPCR experiments throughout the text?
- Figure 2 panels c and e. It is difficult to distinguish the different conditions given the choice of colours.
- Because the 3'UTR of IL1 β -mRNA had a different sequence from β actin-mRNA, the binding might include non-specific and sequence-specific interactions. Maybe good to add 'in vitro'.
- I might be mistaken but I have the impression that the number of cells analysed by microscopy included in the bar plots in Figure 2 and 3 is very small (4-10 cells per condition). Since in a given experiment the number of cells is expected to be high, I don't understand why authors do not include more cells to exclude real signal from artefacts. The fact that data is consistent throughout the figures indicate the this is not the case, but it would be important to comment on this.
- Interestingly the IL1 β -stop-mRNA clearly transported to the nucleus (Fig 3a). Why authors didn't include an illustrative image of this construct, which I think is critical for the interpretation of the results.
- To determine the effect of the ZC3H12D protein, TCM was divided into two parts, and these samples were passed through anti-FLAG-beads, with or without the ZC3H12D-FLAG tag protein. The nex-RNA was isolated from each sample independently. The first nex-RNA was expected to have less IL1 β -mRNA than the second because it was captured in the ZC3H12D-FLAG-beads.

This isn't clear.

Reviewer #2 (Remarks to the Author):

The authors have done an excellent job of extensively addressing the reviewers' concerns. I now wholeheartedly recommend publication of their study.

Thank you for the reviewer's further comments. We have responded to them and appreciate the reviewer's idea to use stop-codon-inserted IL1 β -RNA to demonstrate that our observations were translation-independent.

For reviewer 1

Comment 1: Figure 1d. There is some ZC3H12D in ZC3H12D KO cells. Can authors comment on this? It would be nice to have error bars and t test analysis to see if the differences are significant.

Answer) In the flowcytometric analyses, we used anti-ZC3H12D antibody to stain spleen cells derived from tumor-conditioned medium (TCM)-stimulated mice. To set a gate to isolate ZC3H12D⁺ cells, an isotype control antibody-staining data was used. Our results showed that ZC3H12D⁺ cells were clearly separated from ZC3H12D⁻ cells. Nevertheless, small number of non-specific signals appeared in the gated region. Fc block treatment reduced the non-specific signal, but trace amount of signals remained in the gated region. We added sentence in the legend shown below.

ZC3H12D signals in *Zc3h12d*^{-/-} mouse leukocytes were due to nonspecific antibody binding. (page 39 in the revised text)

Comment 2: RT-qPCR experiments. Can authors add error bars and statistics to all RT-qPCR experiments throughout the text?

Answer) In the previous manuscript, Figure 2b showed simple bar graph depicting mean values for two biologically independent experiments. In the revised manuscript, we added the plot data. In addition, data used in the first submission were displayed in Figure S9. Although they lack β actin and gapdh data, we consider that the data help to show enrichment of IL1 β -mRNA by ZC3H12D pull-down. In Figure S9, we showed Figure 1d, 1h, 1i, 1j, and 5f data obtained from biologically independent preparations. In the Figure 1i dataset, the most important point is that extra-exosomal IL-1 β -mRNA was increase in Lu-TCM compared with Lu-CM. This point was reproduced in the repeated experiment although the absolute values were different from the initial results.

Because an increase of β actin was observed in the repeated experiment, we would like to remove one sentence describing that β actin scarcely increased in the TCM condition. Similarly, in the Figure 1j dataset, the most important point is that IL-1 β -mRNA was captured by ZC3H12D protein beads. This point was reproduced in the repeated experiment, but the absolute values were different from the initial results. We also added note in the figure legend for Fig. 1, 2, 3, and 5, and added sentences in supplementary figure 9 shown below.

Repeated experiment data for Figs. 1d and h to j, 2b, 3a, and 5f. These data show the key points (IL-1 β in Fig. 1i and j and 2b) were reproduced.

Comment 3: Figure 2 panels c and e. It is difficult to distinguish the different conditions given the choice of colours.

Answer) To increase the visibility of the data, FCS data for ZC alone were depicted by dotted lines (2c and e), ZC+IL1 β (3'UTR) 1 nM data was removed (2e), and line colors were changed (2e).

Comment 4- Because the 3'UTR of IL1 β -mRNA had a different sequence from β actin-mRNA, the binding might include non-specific and sequence-specific interactions.

Maybe good to add 'in vitro'.

Answer) We added 'in vitro' in the text shown below.

Because the 3'UTR of *IL1 β -mRNA* had a different sequence from *β actin-mRNA*, the binding might include nonspecific and sequence-specific interactions *in vitro*.

(page 7 in the revised text)

Comment 5- I might be mistaken but I have the impression that the number of cells analysed by microscopy included in the bar plots in Figure 2 and 3 is very small (4-10 cells per condition). Since in a given experiment the number of cells is expected to be high, I don't understand why authors do not include more cells to exclude real signal

from artefacts. The fact that data is consistent throughout the figures indicate the this is not the case, but it would be important to comment on this.

Answer) Thank you for this comment. We consider that we should explain more about the method, regarding with Figure 3a, for the image data acquisition. After application of FITC-labeled RNAs, we stopped the uptake using PFA fixation, and immediately started to obtain the fluorescent signals in nucleus using confocal microscopy. As shown in Methods section, we first detected a cell and decided top and bottom positions of the nucleus in the cell. Then, we obtained 15 continuously sliced images using scan mode. These sliced images were combined for the 3D image re-construction. Thus, to make 6 cell images we accumulated $6 \times 15 = 90$ images. In the graph, 6 data points were plotted per group. This number may look small as cell research data, but the data acquisition process is more laborious than ordinary cell research. The reason why we took this complicated method to obtain the RNA uptake data that labeled RNA emits only weak signals. In this experiment, to minimize the effect of FITC-labeling to the structure of RNA, we reduced the amount of labeled nucleotides in the RNA syntheses as low as possible. Therefore, fluorescence signals were so small that slow scan mode was absolutely required to reduce the noise level, it took about 10 min to obtain one cell data. Thus, it took a total of 5 hrs to obtain 5-6 cells data per sample from 5 samples (non-labeled IL1 β , β actin RNA, gapdh, RNA, IL1 β RNA, and IL1 β -stop RNA). To make accurate comparisons among the samples, we had to complete the data acquisition at least within two days. Therefore, cell numbers in each condition in Figure 3a were limited. To confirm reproducibility of the data, we added repeated data for figure 3a, in supplementary Figure 9. We would like to note that IL-1 β RNA uptake was observed in Figure 3a data and repeated experiment (Fig S9). This conclusion is further supported by Figure 5a data, in which IL1 β RNA uptake are also demonstrated. We also added the sentences in Methods section and figure 5 legend shown below.

Uptake of RNA

... For accurate comparisons among samples, image data were acquired within 2 days. Image data was analyzed by Leica Application Suite X (LAS X v3, Leica). All cells were checked with a single image of the central portion of the nucleus to confirm the nuclear RNA uptake.

(page 23 in the revised text)

Repeated experiment data are shown in Fig. S9. (page 41 in the legend in revised text)

Comment 6- Interestingly the IL1 β -stop-mRNA clearly transported to the nucleus (Fig 3a).

Why authors didn't include an illustrative image of this construct, which I think is critical for the interpretation of the results.

Answer) Thank you for this comment. We added the illustrative image in the top of figure 3a.

Comment 7- To determine the effect of the ZC3H12D protein, TCM was divided into two parts, and these samples were passed through anti-FLAG-beads, with or without the ZC3H12D-FLAG tag protein. The nex-RNA was isolated from each sample independently. The first nex-RNA was expected to have less IL1 β -mRNA than the second because it was captured in the ZC3H12D-FLAG-beads.

This isn't clear.

Answer) We changed the sentences as follows:

To confirm the effect of ZC3H12D protein, migration assays were repeated using TCM passed through ZC3H12D-FLAG tag bound on anti-FLAG-beads. In this experiment, TCM passed through ZC3H12D protein-bound beads [designated as ZC3H12D(+)] column and passed through anti-FLAG beads [designated as ZC3H12D(-)] column, used as a control] were prepared. nex-RNA were isolated from both TCMs and used for the migration assay.